# Stretchable glove for accurate and robust hand pose reconstruction based on comprehensive motion data

Myungsun Park[1,2,6], Taejun Park[1,3,6], Soah Park[4], Sohee John Yoon[1,3], Sumin Helen Koo[4] ✉ & Yong-Lae Park[1,3,5] ✉

We propose a compact wearable glove capable of estimating both the finger bone lengths and the joint angles of the wearer with a simple stretch-based sensing mechanism. The soft sensing glove is designed to easily stretch and to be one-size-fits-all, both measuring the size of the hand and estimating the finger joint motions of the thumb, index, and middle fingers. The system was calibrated and evaluated using comprehensive hand motion data that reflect the extensive range of natural human hand motions and various anatomical structures. The data were collected with a custom motion-capture setup and transformed into the joint angles through our post-processing method. The glove system is capable of reconstructing arbitrary and even unconventional hand poses with accuracy and robustness, confirmed by evaluations on the estimation of bone lengths (mean error: 2.1 mm), joint angles (mean error: 4.16°), and fingertip positions (mean 3D error: 4.02 mm), and on overall hand pose reconstructions in various applications. The proposed glove allows us to take advantage of the dexterity of the human hand with potential applications, including but not limited to teleoperation of anthropomorphic robot hands or surgical robots, virtual and augmented reality, and collection of human motion data.

Dexterity and versatility, clear distinctions of the human hand from those of animals, allow myriad functions ranging from grasping[1,2] to in-hand manipulation and even to means of communication using hand gestures[3,4]. Tracking and reconstruction of the hand articulation is, therefore, a popular topic of research with numerous applications, including robotics[5], healthcare[6], gaming[7], virtual and augmented reality[8]. However, there have been many challenges in estimating hand motions, as the human hand is a complex body part with many degrees of freedom (DoFs) and shows large variations between individuals. Many prior studies have demonstrated classification and reconstruction of hand poses[9–13], albeit within a limited scope, providing partial evaluations on the hand motion tracking involving only part of the full range of finger motion, restricted hand positions, or simple applications limited to a small number of general hand poses. Such performances may not be sufficient for applications, such as teleoperation of surgical robots[14], control of anthropomorphic robot hands[15,16], and clinical analysis[17], which demand precise localization of the fingertips (e.g., 1 mm fiducial localization error of da Vinci system[18]) and detailed posture of the hand (e.g., Kapandji score, a clinical assessment of the thumb opposition on a scale of 0 to 10[19]). To replicate these hand motions, it is essential to identify the hand configuration, which can be characterized by both the kinematic structure and the joint motions[20]. Consequently, the accurate quantification and measurement of these two components hold significant importance.

[1]Department of Mechanical Engineering, Seoul National University, Seoul 08826, South Korea. [2]Department of Mechanical and Aerospace Engineering, University of California San Diego, La Jolla, CA 92093, USA. [3]Institute of Advanced Machines and Design, Seoul National University, Seoul 08826, South Korea. [4]Department of Clothing and Textiles, Yonsei University, Seoul 03722, South Korea. [5]Institute of Engineering Research, Seoul National University, Seoul 08826, South Korea. [6]These authors contributed equally: Myungsun Park, Taejun Park. ✉e-mail: smkoo1@yonsei.ac.kr; ylpark@snu.ac.kr

To quantify the joint motion, prior studies have employed strain/bending sensors, encoders, or inertial measurement units to directly measure the joint angles or the bone rotations[21–23]. Others have also utilized Triboelectric nanogenerators to detect charge generation with hand motions, or actuating-sensing bone conduction methods to analyze the mechanical waves traveling through the bones[24,25]. As a counterpart, vision or magnetic sensors have been used to track specific features of the hand, such as fingertips[11,26]. The measured quantities from either of these approaches then can be used to reconstruct the hand configuration through forward kinematic (FK) or inverse kinematic (IK) models, respectively, combined with the kinematic structures. To identify the kinematic structures, vision-based techniques[27,28] have been widely used. However, these methods require highly controlled setups and are susceptible to the orientation of the cameras and occlusions, limiting the wearer's free hand motion in various environments. Other studies have manually measured the bone lengths[29,30] but the result of this method can be highly dependent on the expertise of the examiner. A more simplified approach has also been adopted by applying the average bone lengths to all users rather than selecting personalized parameters[16], which however resulted in large errors in FK or provided multiple or no solution in IK[31,32]. To mitigate these issues, some studies have proposed the application of complex kinematic constraints to the hand motions[33,34]. However, this approach requires heavy computation and may inadvertently exclude potential solutions. While some of these issues can be addressed through integration of multiple sensing mechanisms, this approach often results in bulky and restrictive systems with reduced wearability[35]. There have also been attempts to reduce the complexity of the hardware and enhance the wearability of the hand motion sensing system, focusing on differentiating a limited number of hand poses rather than precise estimations of the hand configuration parameters nor its reconstruction[25]. Therefore, there still exists a significant demand for a hand tracking system that overcomes these limitations in terms of accuracy (i.e., hand reconstruction error[36]), robustness (i.e., ability to reconstruct arbitrary poses[35]), and wearability (physical comfort and compact form factor).

In this research, we propose a wearable glove capable of real-time estimation of the wearer's finger bone lengths and the joint angles with high accuracy, all achieved through a single sensing mechanism. The glove system is calibrated and evaluated using comprehensive hand motion data that reflect a wide range of natural human hand motions. Furthermore, our system is able to reconstruct complex and unconventional poses taking advantage of the simple yet robust FK reconstruction method, making it suitable for various applications. The glove measures the lengths and the motions of the thumb, the index, and the middle fingers. However, the same sensing mechanism may be extended to measure the rest of all five fingers that share the same kinematic structure. Below, we outline the key features and the contributions of our glove system.

First, the system uses a single strain sensing mechanism to estimate both the wearer's kinematic parameters (i.e., bone lengths) and joint motions (i.e., joint angles) that define the hand pose. This is enabled by the highly stretchable and flexible characteristic of the soft strain sensors used in this study. The soft strain sensors can be pre-stretched to fit the hand of the wearer and then be further deformed easily by the joint movements. Therefore, the sensor signals simultaneously provide the information on both the bone lengths and the joint rotations. As a result, once the glove is worn, it can immediately start tracking and reconstructing the wearer's hand without the need of any prior measurements or calibration for personalization. The compact design of our system also demonstrates a consistent performance, unaffected by various factors, including position, orientation, or environmental conditions, making itself a truly wearable system.

Second, the calibration and evaluation of our glove system rely on a comprehensive set of ground truth data, which represents the unconstrained and diverse hand motions expressed in the universal joint coordinate. This differentiates our system from the previous studies that evaluated their systems only for limited experimental conditions and users. For example, the strain-based tracking system was evaluated with the data obtained from the static motions of a hand replica[9], and the vision-based system was tested only for restricted hand motions confined to a customized stage[37]. Another approach with vision used datasets of fingertip positions, highly dependent on the specific kinematic structure, and thus may not be universally applicable to all hand configurations[38]. In contrast, we collected a reference dataset that captures the full ranges of all DoFs in various free-hand motions, expressed with joint angles. Using a customized motion capture setup and marker layout, we captured the pure rotation of each finger bone. Since joint angles are universal representation of the human hand motion that does not depend on anatomical variation between individuals[4], we developed a post-processing method that extracts the joint angles from the collected data. When trained and tested with the comprehensive ground truth data, our system demonstrated enhanced performance over a wider range of wearers and diverse hand poses.

Finally, using the estimated kinematic parameters and joint motions, expressed in the form of the bone lengths and the joint angles, the system adopts the FK method to reconstruct the hand poses[39]. The FK method does not require a complex set of kinematic constraints often required for IK approaches[40] and the reconstruction is therefore not limited to the scope of the constraints or the training dataset. Taking advantage of this simplicity and efficiency, we demonstrate that our system is able to successfully reproduce the entire range of motions of the fingers, including various degrees of thumb oppositions, complex interactions, and unconventional poses of the fingers through multiple applications.

## Results
### Overview of the glove hardware design
Glove-type soft wearable sensors have an advantage in hand motion sensing over other types of sensing methods as they are not limited by occlusion, strictly controlled environment[11,41], or bulky and rigid components[42,43]. Here, we present a wearable sensing glove that provides accurate and robust estimation of the human's hand kinematic structure and joint motion, all while ensuring comfortable wearability. The proposed sensing glove (Fig. 1a) consists of three key components: a liquid-metal soft sensor layer characterized by its high sensitivity and flexibility to accommodate hands of all sizes; a custom-designed textile glove interface designed based on the human hand anatomy and kinematics; and a watch-type circuit board for signal acquisition and conditioning (further details provided in the Methods section), securely enclosed and worn on the wrist for wireless transmission of sensor measurements. Since the kinematic structure of the ring and the little finger joints resemble those of the index and the middle fingers, we only test the sensing performance for estimating the lengths and motions of the thumb, the index, and the middle fingers to prove the concept of our study. This was considered to be adequate in validating the performance of the proposed system.

### Soft and sensitive liquid metal-based sensor
The top sensing layer is composed of a silicone substrate (Ecoflex 00-30, Smooth-On, 100% modulus: 69 kPa, elongation at break: 900%) embedded with nine traces of eutectic Gallium–Indium (eGaIn), a room-temperature liquid metal[44]. The soft and stretchable substrate can accommodate a wide range of movements and conform to irregular contours of the human hand[45,46]. The liquid-metal traces are used to measure deformations of the hand as it conforms to the geometrical

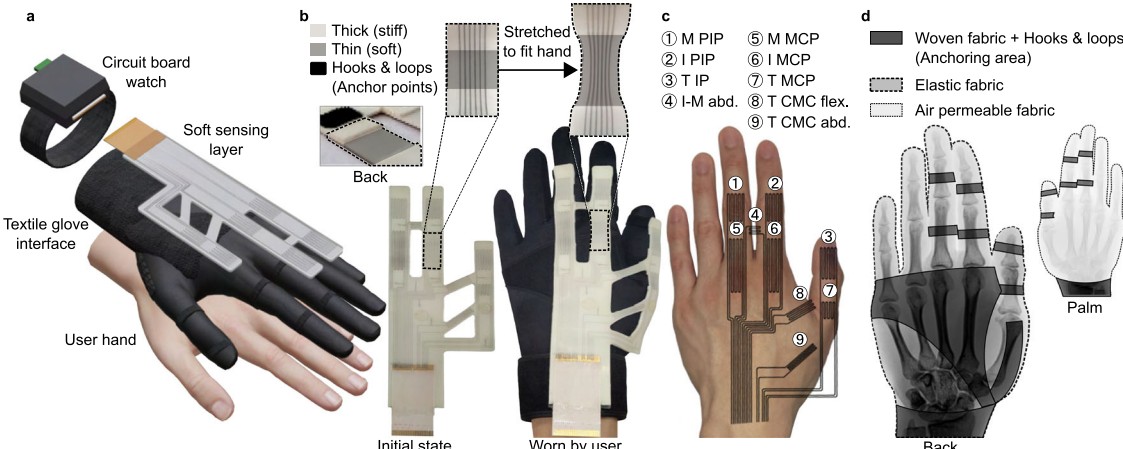

**Fig. 1 | Glove hardware design. a** Schematic diagram of the proposed glove system hardware comprising the soft sensing layer, the textile glove interface, and the watch-type circuit board shown with the hand of the wearer. **b** Image of the initial state of the soft sensing layer (left) and its pre-stretched state (right) when anchored onto a medium-sized hand; the inset images show the multi-stiffness substrate structure and its characteristic deformation when stretched. **c** Placement of the liquid-metal sensor traces with respect to the hand. M middle, I index, T thumb, PIP proximal interphalangeal, IP interphalangeal, MCP metacarpophalangeal, CMC carpometacarpal, flex flexion, abd abduction. **d** Design of the textile glove interface composed of elastic fabric for unrestricted hand motion and hook-and-loop textile for anchoring; their positions are shown with respect to the finger bones of the wearer.

changes of the soft substrate during stretching, resulting in changes in resistance[47].

The sensing layer and the layout of the traces were strategically designed to simultaneously measure both the finger bone lengths and the joint motions. The dimensions of the sensing layer were intentionally designed to be smaller than the size of the hand, and when initially donned by the wearer, it must be stretched to adequately encompass all three fingers (Fig. 1b). This initial stretch of the sensors provides a preliminary estimation of the hand size and finger bone lengths. The positions and the orientations of the nine eGaIn traces were also designed to measure strains caused by motions of the 10 DoFs of the seven joints listed in the Methods section (Fig. 1c). During hand motions, the eGaIn traces of the sensing layer are further stretched or relaxed resulting in changes in resistance, providing information on the dynamic joint angles as well as the finger bone lengths. Further details on the estimation method will be discussed in the following subsections.

The mechanical structure of the sensing layer was designed to increase the resolution of liquid-metal soft strain sensors, resulting in accurate measurement of the bone lengths and the joint angle. One approach to increase the sensitivity of the liquid-metal strain sensors is to design the substrate with a combination of multiple elastomer strips with different stiffnesses[48]. These sensors are also known for their reliability in prolonged and repeated loading cycles, as demonstrated in previous studies[48], which overcomes the structural limitation often faced by soft, stretchable sensors. We thus employed this approach, creating the stiffness variations by altering the thickness of the substrate with a single material, as shown in Fig. 1b. This method not only enhanced the structural integrity of the sensor but also improved the sensitivity (gauge factor: 3.4).

Hook-and-loop (Velcro®) patches were bonded to the bottom surface of the sensing layer to anchor the layer onto the textile glove interface. By securing the layer onto the middle of each finger bone, each sensor was positioned directly over each finger joint (Fig. 1d). This anchoring method serves multiple crucial purposes, contributing to the reliability and repeatability of the glove system. Firstly, the anchor points serve as fixed reference points, not allowing rotation or displacement and ensuring all the sensors are subjected to uniaxial tension only. Consequently, we can safely assume that the sensor signals are only affected by the strains resulting from the joint rotations. Secondly, the anchoring method not only enforces an injective

relationship between the sensor signals and the joint angles but also minimizes coupling between the sensor signals. This means that every set of the sensor signals is unique to a combination of the joint angles and the bone lengths, representing a distinct hand configuration. Finally, the anchor points also provide a consistent and repeatable method for attaching the sensing layer. This is particularly important as the system must fit and track different hands with various sizes. The placement of the anchor points at the midpoint of the bones is easily identifiable for all hands, allowing for the consistent attachment of the sensing layer to the hand with minimal deviations in the sensor signals across repeated trials. Fabrication of the soft sensing layer is described in the Methods section and Supplementary Fig. 1, and the sensor signals during repeated trials are shown in Supplementary Fig. 2.

## Custom textile glove interface

A customized textile glove was used as an interface between the sensing layer and the wearer's hand (Fig. 1a). This interface serves the purpose of providing robust anchor points that effectively decouple the joint rotation of each finger from those of the other joints while minimizing any physical interference or discomfort caused by sensors. Figure 1d depicts an example X-ray image of a male human hand. In order to allow natural finger movements without restrictions, the fabric surrounding the finger and the thumb joints must allow large deformations. Simultaneously, other parts, such as the finger bones located between the joints or on the back of the hand must provide sturdy anchoring points that maintain their positions relative to the wearer's hand. Therefore, elastic and inextensible fabrics were selected and patterned to provide a tight fit over the hand during dynamic motions, facilitating comfortable joint articulation. This approach also guarantees that all deformations are solely induced by the finger articulations. Further details are discussed in the Methods section and Supplementary Tables 1 and 2. Supplementary Video 1 (highlights provided in Supplementary Information) is provided to show the anchoring process of the sensing layer, and also various hand motions that cover the full range of motion of the fingers, in comparison to a bare hand.

## Overview of collecting, modeling, and post-processing for comprehensive dataset

The fundamental goal of our hand tracking system is to accurately quantify and measure the wearer's kinematic structure and the joint

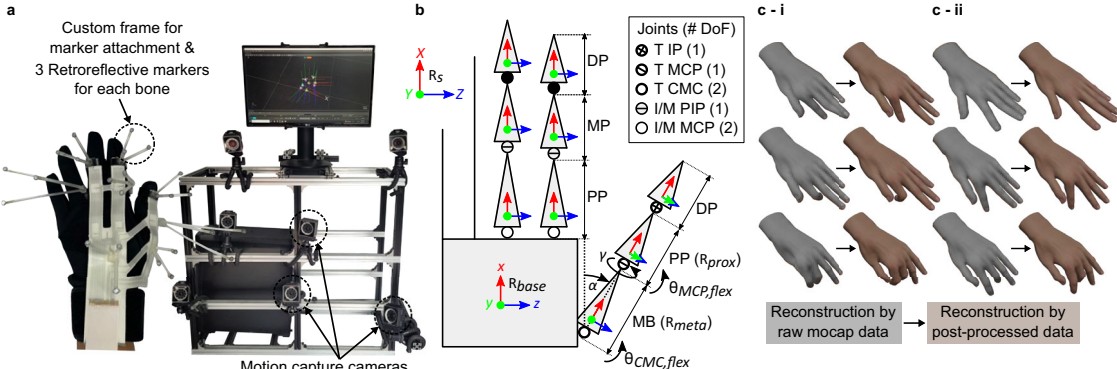

**Fig. 2 | Collecting, modeling, and post-processing of the comprehensive hand motion dataset. a** Motion-capture setup for collecting hand motion data with custom retro-reflective marker frames to define the rigid body of each finger bone (left), and custom motion-capture camera stand (right). **b** Forward kinematic hand model at zero-pose showing the local frame of each finger bone, the joint positions, and the parameters defining the wearer's hand anatomy. $R_s$, $R_{base}$, $R_{meta}$, and $R_{prox}$ represent the global motion-capture frame, the local hand base, metacarpal bone, and proximal phalanx frames, respectively. $\alpha$, $\gamma$, and $\theta$ represent abduction offset, roll offset, and joint angles. M middle, I index, T thumb, DP distal phalanx, MP middle phalanx, PP proximal phalanx, MB metacarpal bone. **c** Reconstruction of hand poses using the raw motion-capture data (grayscale) and the post-processed data (colored).

motion. To meet this goal, we must calibrate and evaluate the system using comprehensive training data that reflects the vast range of free hand motions. In the following subsections, we describe the procedures employed to acquire comprehensive ground truth data of the hand motions. We first present the motion-capture setup utilized for collecting raw hand motion data. Subsequently, we explain the details of the FK hand model on which our joint angle processing and hand reconstruction methods are based. Lastly, we discuss the post-processing method to extract the joint angles from the raw motion-capture data. Since these joint angles are not dependent on the variations in hand anatomy, this method allows the captured data from any subjects to be applicable to the general FK model to represent a wide range of wearers.

## Motion-capture setup to collect comprehensive motion data

Figure 2a shows the experimental setting for collecting diverse and unconstrained hand motion data which were used to calibrate and to evaluate our system. A commercialized motion-capture system (Opti-Track, NaturalPoint) was used to track the rotation of each bone. In various prior studies[49], retroreflective markers were attached to the finger joints, and their positions were tracked to calculate rotation of the finger bones through the difference between two consecutive vectors, formed by neighboring markers. While this method is simple, it is highly susceptible to uncertainty in localizing the exact joint positions on the deformable skin, particularly during motion[50]. Instead, we used three markers (3 mm diameter) to define each bone as a rigid body to directly measure its rotation, which did not require precise positioning of the markers[49,51]. Since the human hand is a very small target[52,53] with a large range of motion, it is difficult to resolve the neighboring markers and avoid self-occlusions if the markers are directly attached to the fingers[20,54]. Therefore, we designed custom frames that protruded the markers outwards to provide enough spaces (>30 mm) between the markers and to prevent self-occlusion during the motion. These frames were attached to the anchor positions of the glove (Fig. 1d) to ensure that the movements of the frames were driven only by the rotation of the corresponding bone.

## Forward kinematic hand model for various hand anatomy

Figure 2b displays the FK hand model in its zero pose that we used to reconstruct the hand poses. In this model, we quantify the hand configuration using the bone lengths, the joint angles, and the anatomical parameters ($\alpha$, $\gamma$). Detailed description of the variables, the zero pose, and the local frames of the kinematic model displayed in the figure are

provided in the Methods section. Since our metacarpophalangeal (MCP) and the Interphalangeal (IP) joints move with one DoF, by assigning the local frames in a specific orientation with the abduction offset $\alpha^*_{abd}$ and the roll offset $\gamma^*_{roll}$, the rotation of the proximal phalange and the distal phalange can be expressed using a single flexion angle. These specific offset values, which will be referred to as *true offsets* hereafter, depend on the anatomy of the individual and can be identified by analyzing the rotations of the bones during hand motion.

## Post-processing for identifying hand anatomy and extracting joint angles

We identified the *true offsets* from the raw bone rotations obtained through the motion capture system by expressing the rotation matrices of thumb metacarpal bone (MB) ($\mathbf{R}_{meta}(\theta_{CMC,flex})$) and the thumb proximal phalange ($\mathbf{R}_{prox}(\theta_{MCP,flex})$) in the global frame, by assigning arbitrary offsets. We then determined the *true offsets* as the optimum offset values that align the axes of flexion motions of the bones. Detailed description of the identification method is provided in the Methods section.

By assigning the local frames on every bone and phalange based on the identified offsets, relative rotations between successive rigid bodies were calculated. For 1-DoF joints (PIP of the index and the middle finger, IP and MCP of the thumb), the angles in axis-angle representation of the relative rotations were regarded as the flexion angles[55]. For 2-DoF joints (MCP of the index and the middle finger, and CMC of the thumb), we interpreted the pitch and the yaw angles of the relative rotations as the flexion and the abduction angles since the *true offsets* result in negligible roll angles.

We confirmed that the reconstructed hand poses using the hand model from Fig. 2b and the post-processed joint angles were able to accurately represent the target hand poses, as shown in Fig. 2c. Here, the motion of the ring and little fingers were not directly measured by the motion capture system. Their joint angles were assumed to be two-thirds and two-fifths of the corresponding joint angles of the middle finger, respectively, based on the concept of kinematic synergies of hand grasps[56,57] for visualization purposes. The poses on the left (grayscale) were reconstructed by direct application of the 21-DoF raw motion-captured bone rotations to the finger bones. The poses on the right (colored) were reconstructed by inputting the processed joint angles through the method described above into the 10-DoF hand model in Fig. 2b. It is apparent that the proposed method is able to reproduce the original poses by correcting unnatural motions (Fig. 2c–i) that result from the redundant DoFs.

**Table 1 | Mean and standard deviation of the absolute errors in initial bone lengths estimation collected from 13 subjects, in length (mm) and normalized value (%) by the true bone lengths**

| Absolute error in initial bone lengths estimation (mm) | | | | | | | | | |
|---|---|---|---|---|---|---|---|---|---|
| Bones | I DP | I MP | I PP | M DP | M MP | M PP | T DP | T PP | T MB | Average |
| Mean | 1.70 | 1.68 | 2.79 | 2.15 | 1.62 | 2.77 | 1.75 | 2.18 | 1.93 | 2.06 |
| Standard deviation | 1.21 | 1.23 | 1.85 | 1.13 | 1.26 | 2.65 | 1.14 | 2.02 | 1.68 | 1.57 |
| Normalized absolute error in initial bone lengths estimation (%) | | | | | | | | | |
| Bones | I DP | I MP | I PP | M DP | M MP | M PP | T DP | T PP | T MB | Average |
| Mean | 7.88 | 7.67 | 7.57 | 9.20 | 6.29 | 6.69 | 6.48 | 5.75 | 4.88 | 6.93 |
| Standard deviation | 5.97 | 5.99 | 5.21 | 5.06 | 4.89 | 6.93 | 4.88 | 4.48 | 4.25 | 5.30 |

*I* index, *M* middle, *T* thumb, *DP* distal phalanx, *MP* middle phalanx, *PP* proximal phalanx, *MB* metacarpal bone. Source data are provided in the Source Data file.

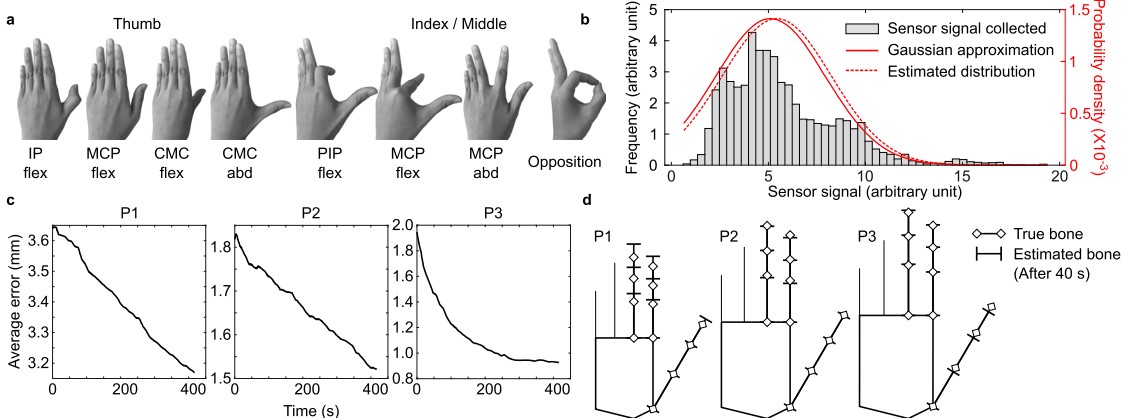

**Fig. 3 | Real-time refinement of bone lengths using signal distributions.**
**a** Collection of hand poses conducted for the refinement of bone lengths estimation. All motions were restricted to these eight poses. IP interphalangeal, MCP metacarpophalangeal, CMC carpometacarpal, PIP proximal interphalangeal, flex flexion, abd abduction. **b** Histogram of the sensor signals of I-M abd. in Fig. 1c collected over the hand motion test; its Gaussian approximation defined by the true mean and variance of the histogram, and the estimated distribution defined by the mean and covariance estimated from the subject's bone lengths. **c** Plot of average estimation errors of 10 bones with respect to time of continuous hand motions, shown for three participants (P1–P3). **d** Graphical representation of the estimated and true bone lengths of the three participants (P1–P3). Source data are provided in the Source Data file.

## Overview of bone lengths and joint angles estimation

Our glove system takes advantage of the stretchable sensor mechanism to attain information on both the bone lengths and the joint angles. The initial stretch of the sensor when it is worn on the hand, provides the initial estimation of the wearer's bone lengths. Afterwards, the sensor signals obtained during dynamic hand motions are used to refine this estimation as well as estimate the joint angles. The system is calibrated and evaluated for general wearers using the comprehensive ground truth data discussed in the previous subsections.

## Initial estimation of bone lengths using initial sensor signal

The underlying assumption for deriving the initial estimation of the bone lengths from the initial sensor signal is that a unique set of sensor signals is generated by each different hand anatomy thanks to the design of the consistent anchor points in Fig. 1d. We collected data from thirteen subjects (nine males and four females) to test this assumption and characterize the estimation model. Details on the testing procedure and characterization method will be described in the Methods section and Supplementary Note 1, Supplementary Fig. 2, and Supplementary Fig. 3. The results of the initial bone lengths estimation are summarized in Table 1 (absolute estimation error in mm and normalized absolute estimation error by the bone lengths in %). The mean absolute errors averaged over all bones was 2.06 mm, ~6.9% of the target bone lengths. The results of the initial bone lengths estimation of all 13 subjects are presented in Supplementary Table 3.

## Real-time refinement of bone lengths using signal distributions

Once the wearer starts moving the fingers, the change in the sensor signals form a unique distribution depending on the hand size, which can be used to further refine the initial estimations made in the previous subsection. Based on the Bayes rule[58], we refined the bone lengths using the subsequent sensor signals collected from the thirteen subjects while they were moving their fingers. The hand motions were constrained to the poses illustrated in Fig. 3a to simplify the problem. Detailed descriptions of the experiments, fitting of the sensor distributions, and the refinement method are provided in Method section. An example of the fitted likelihood of the sensor signals is presented in Fig. 3b. The detailed description of the figure is also provided in the Method section.

However, the Gaussian assumption for the distribution of sensor signals and the following analytical solutions, as presented in the Methods section, may be invalid for arbitrary motions not shown in Fig. 3a. To release this constraint, i.e., to allow the wearer to move their hand more freely while updating the bone lengths estimation, the design of the likelihood function may have to be non-Gaussian and the posterior should be calculated using different methods[59,60], which we leave for future work.

Figure 3c, d exemplify how the update algorithm refines the bone length estimation in three subjects (two males, one female). For all three subjects, the average estimation error (Fig. 3c) of the ten bone lengths continuously decreased with time as the subjects moved their hands. After 400 s, the average error was reduced by 12.9%, 16.3%, and

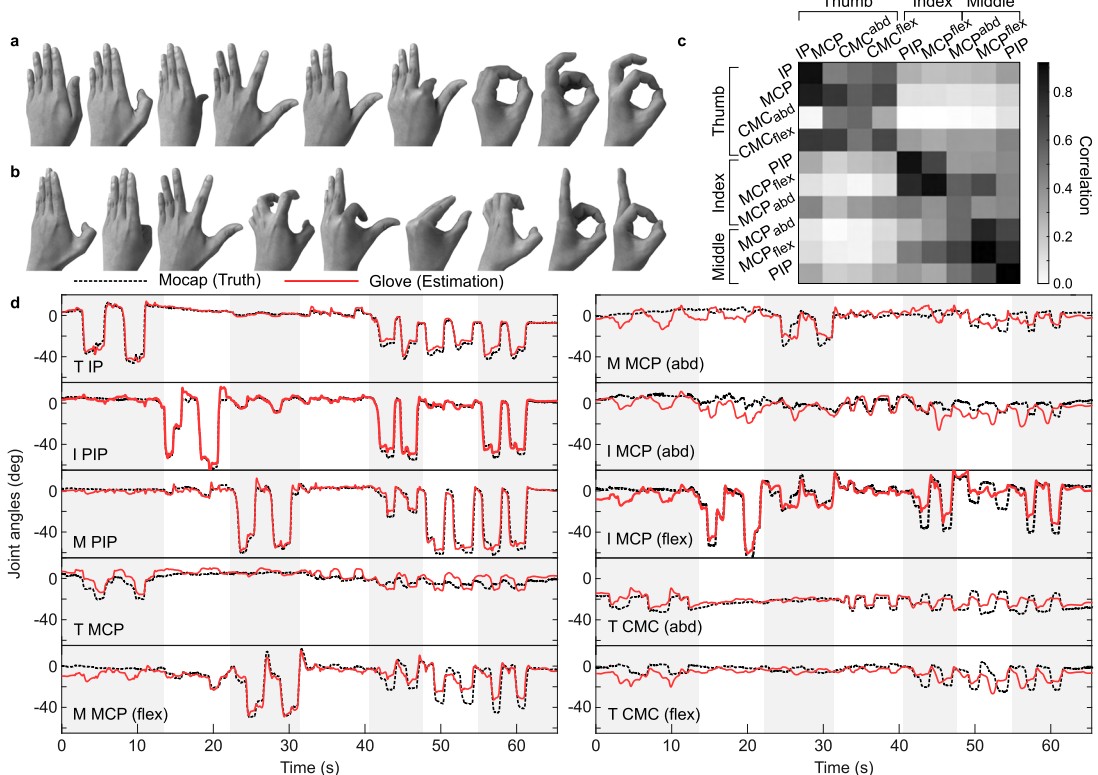

**Fig. 4 | Real-time estimation of joint angles.** Collection of hand poses conducted during the calibration (**a**) and the evaluation (**b**) for the joint angle estimation. **c** Graphical representation of the correlation between the joint angles (vertical) and the sensor signals (horizontal) in calibration dataset. **d** Plot of the time trajectories of the estimated and true joint angles for the large hand subject in the *self-calibration* group. T Thumb, I Index, M Middle, IP interphalangeal, PIP proximal interphalangeal, MCP metacarpophalangeal, CMC carpometacarpal, flex flexion, abd abduction. Source data are provided in the Source Data file.

8.3% for each subject from their initial estimates. This refinement produced a more accurate estimation of the bone structure that better resembled the true hand, as graphically depicted in Fig. 3d. The results of the refined bone lengths estimation of five subjects are summarized in Supplementary Table 4.

## Real-time estimation of joint angles

To calibrate and test the joint angle estimation model, we first collected a comprehensive hand motion dataset. The motion data were collected from two groups of subjects: (1) *self-calibration* group (one male, one female) provided the calibration and the test data, and (2) *non-calibration* group (one male, one female) only provided the test data. During the experiment, the subjects performed diverse and unconstrained hand motions that utilized the full range of motion of each finger joint. The poses are partially shown in Fig. 4a, b.

Figure 4c visualizes the correlation between the joint angles and the sensor signals in the calibration data. This map presents dominant diagonal elements, which represent a strong correlation between the joint angle of each finger and its corresponding sensor signal (Fig. 1c), as well as significant off-diagonal elements attributed to the synergetic nature of the hand motions[56,57] and mechanical coupling between the sensors. Based on these relationships between the sensor and the joint angles, a system model with a state ($\mathbf{x}_k \in \mathbb{R}^{10 \times 6}$) of the joint angles captured in the previous 0.1 s (series of 6 data points at 60 Hz) and the sensor observation ($\mathbf{y}_k \in \mathbb{R}^9$) was defined as

$$\begin{aligned}
\mathbf{x}_{k+1} &= \mathbf{A}\mathbf{x}_k + \mathbf{w}_k, \\
\mathbf{y}_k &= \mathbf{H}\mathbf{x}_k + \mathbf{v}_k, \\
\mathbf{w}_k &\sim N(0, \mathbf{Q}), \mathbf{v}_k \sim N(0, \mathbf{R})
\end{aligned} \tag{1}$$

Details on implementing this linear model for the joint angle estimation will be described in the Methods section. The bottom line is that, in this study, we categorized the subjects into small and large hands and separately calibrated to compensate for different levels of pre-stretching and to achieve accurate estimation.

The results of joint angle estimation (the mean absolute errors of each joint) are summarized in Table 2. Figure 4d shows the result of the large hand subject in the *self-calibration* group, comparing the time trajectories of the estimated and the true (motion-captured) joint angles. For all hand sizes, and for both *self-calibrated* and *non-calibrated* subjects, the average angle estimation outperformed the off-the-shelf hand tracking systems evaluated in[16]. Although the errors in the *non-calibration* group were larger than those of the *self-calibration* group, the calibration method can be further refined with more dataset of various hand sizes, and the system may be universally adopted for a large range of arbitrary wearers with a high accuracy.

## Fingertip position estimation

First, we conducted a quantitative analysis of the hand pose reconstruction by predicting the fingertip positions. Retroreflective markers were attached to the fingertips, and their positions were tracked while the wearer moved the hand with the glove. The test setup and the procedures are described in Fig. 5a and the Methods section. Figure 5b and Fig. 5c-i compare the reconstructed and the ground truth (motion-captured) 3D fingertip trajectories for the small and the large hands, respectively, and Fig. 5c-ii depicts the time trajectory results for the large hand. The mean absolute errors of the fingertip tracking test are presented in Table 3.

Overall, the reconstructed trajectories coincide with the ground truth. However, it underestimated the range of the motions and the result of the smaller hand also showed distorted trajectories in space.

**Table 2 | Mean absolute errors of the finger joint angles for the subjects in the *self-calibration* group and *non-calibration* group**

| Subject | | Mean absolute estimation error (degree) | | | | | | | | | | | | | |
|---|---|---|---|---|---|---|---|---|---|---|---|---|---|---|---|
| | | T IP flex. | I PIP flex. | M PIP flex. | T MCP flex. | M MCP flex. | M MCP abd. | I MCP abd. | I MCP flex. | T CMC abd. | T CMC flex. | Average |
| *Self-calibration* | Small hand | 2.10 | 2.05 | 2.27 | 11.83 | 4.51 | 2.23 | 3.98 | 4.62 | 1.57 | 3.93 | 3.91 |
| | Large hand | 1.85 | 2.17 | 2.68 | 4.15 | 4.61 | 5.23 | 6.16 | 5.47 | 3.79 | 5.48 | 4.16 |
| *Non-calibration* | Small hand | 1.77 | 3.01 | 4.85 | 3.00 | 8.88 | 5.75 | 3.42 | 0.47 | 15.08 | 0.62 | 4.69 |
| | Large hand | 5.44 | 4.73 | 3.94 | 3.96 | 11.93 | 7.57 | 8.67 | 13.11 | 3.13 | 8.93 | 7.14 |

Source data are provided in the Source Data file.

These errors were most likely generated by the combined effect of the errors in the joint angles, the bone lengths, and the hand anatomy ($\alpha_{abd}$ and $\gamma_{roll}$). The underestimation of the fingertip positions was a result of the underestimated joint angles, as shown in Fig. 4d as well as the errors in the bone lengths. The distortion in the thumb motions is attributed to the errors in the abduction and the roll offset that decouple the abduction and the flexion motions.

Nevertheless, the mean absolute 3D position errors were 3.24 mm and 4.02 mm for the small and large hands, respectively, lower than a half of the smallest reported error of among commercially available hand tracking systems[17]. These 3D errors were both ~16% of the side length of the workspaces, which suggests the performance of the fingertip position reconstruction is consistent regardless of the hand sizes.

### Real-time hand pose reconstruction

Qualitative assessment of the proposed hand reconstruction method was conducted by visualizing the result in a virtual environment and comparing it to the real hand. An open-source 3D computer graphics software tool (Blender) was used to create a virtual hand in close proximity to the real human hand (Fig. 6a). The finger bone lengths and the joint angles estimated by the proposed glove system are the inputs to update the length and the rotation angle for each virtual finger bone in real time at 30 Hz. The reconstruction was based on FK, and no additional constraints were applied to the motion of the virtual hand. Since the ring and the little fingers were not directly measured by the glove, their rotation angles were assumed as two-thirds and two-fifths of the middle finger joints, respectively, for visualization purposes. These two fingers were not included in the evaluation of the performance of the glove system, which is a limitation of the glove in its current state since other variations of precision grasps may require independent motions of each finger for example. For a more comprehensive estimation of the full hand motion, the system can be easily extended using additional sensors and the same post-processing method, which we pose as future work.

We first reproduced free hand motions involving articulation of each individual finger, as well as their combined motions. Examples of the reconstructed hand poses are shown in Fig. 6b and Supplementary Video 2 (highlights provided in Supplementary Information). The reconstructed hand poses closely resembled the real hand poses, accurately reproducing the full DoFs of the finger joints. Particularly, the ability to successfully reconstruct contact or proximity of multiple fingertips through FK without any kinematic constraints or tactile sensors highlights the accuracy of the bone lengths and joint angle estimations.

Figure 6c shows the reproduction of various grasping poses (Supplementary Video 2 and highlights in Supplementary Information). The virtual hand closely reflected the unique features of the grasping poses ranging from spherical and tripod grips to precision grips (two-fingered pinching), showing the glove performance unaffected by the surrounding environment or contacts with the object in grasp.

We also conducted a more standardized test to evaluate the accuracy of the glove system in reconstructing opposition of the thumb and its interaction with other fingers using the Kapandji score[19]. However, since the glove system only directly estimates the motions and the lengths of three fingers (thumb, index, and middle finger), and accurate reconstruction of the ring and little fingers are not evaluated in this research, the test was modified so that the thumb makes contact with the DP, MP, and PP bones of the index and the middle fingers only. Therefore, the original scoring metric was not applied to this modified test. The results (Fig. 6d and Supplementary Video 3 and highlights in Supplementary Information) suggest the proposed glove is capable of sensing and reconstructing the opposition of the thumb as well as accurately reconstructing various forms of interactions (contacts) of the fingers.

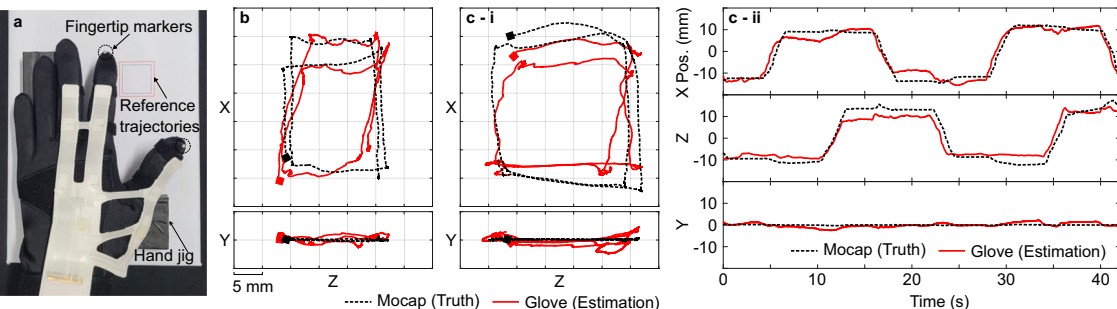

**Fig. 5 | Fingertip position estimation. a** Test setup for prediction of the fingertip position using the bone length and the joint angle estimations. **b** 3D trajectories of the true (motion captured) and the estimated fingertip positions for the small-handed subject. **c** 3D trajectories (**i**) and the time trajectory (**ii**) of the true (motion captured) and estimated fingertip positions for the large-handed subject. Source data are provided in the Source Data file.

## Application 1: number pad typing

To demonstrate the performance and the potential of our sensing glove for use in real-world applications, we first conducted a task of typing a virtual number pad (Fig. 7a, Supplementary Video 4 and highlights in Supplementary Information). Typing, although a very mundane task, requires precise manipulation of the fingers and accurate positioning of the fingertips. Since the fingertips must make contacts with specific locations of the keys, the success of the virtual typing largely depends on accurate estimation of the fingertip position as well as the overall hand configuration that does not cause any unintentional interactions with the keyboard. Since the proposed glove was designed for the left hand of the wearer, we operated a virtual number pad, which can be done with a single hand without controlling the wrist motion. The workspace of the two tasks shown in the demonstration was 80-by-60 mm, which covered the entire area of the number pad, and thereby proved the accuracy of the fingertip position estimation in the large task space.

## Application 2: shadowgraphy

Next, to prove the robustness of our system in complex and unconventional poses, we demonstrated virtual shadowgraphy. Shadowgraphy is the art of creating distinct figures, such as animals or humans, by projecting shadows casted by the performer's hands. This task requires articulate control of the hand, involving very complex, unusual, and unique hand poses that we may not use in our day-to-day lives. Therefore, it is a good example that can highlight the robustness of our FK method that can accurately reconstruct even uncommon and unnatural hand postures using accurate finger bone lengths and joint angles estimations. As shown in Fig. 7b, a set of twin environments (real and virtual) was set up for real-time comparison of the casted shadows (Details in the Methods section).

We demonstrated shadowgraphy of five animals shown in Supplementary Video 5 (highlights provided in Supplementary Information). Some differences between the real and the virtual shadows were apparent. However, such dissimilarities can be attributed to the differences in the relative position of each hand (virtual and real) with respect to its environmental conditions (e.g., light, screen, or camera),

and the differences in hand contour due to the textile glove and the sensing layer. Overall, the proposed system successfully reconstructed complex hand poses to create all five animals.

## Application 3: teleoperation of dexterous robotic hand

We also performed teleoperation of an anthropomorphic robotic hand (Allegro Hand, Wonik Robotics), for precise manipulation of various objects in practical situations (Fig. 7c, Supplementary Video 6 and highlights in Supplementary Information). We carried out three different tasks that required dexterous and precise manipulation. The first was manipulation of a ball on a table, repositioning the ball to a desired location and grasping it. The next task is to control the robotic hand to manipulate a mouse wheel, to scroll through pages of a document. This task required accurate control of the fingertip position so that the finger only rotates the mouse wheel without pressing it and activating a different mouse function. Finally, a volume control knob of an audio interface was manipulated by the robotic hand, which required unique sequences of multiple finger motions to rotate the cylindrical knob in both directions. The results shown in Supplementary Video 6 demonstrate that our glove system can be used to take full advantage of the dexterity of the human hand in various practical scenarios.

# Discussion

## Contributions, limitations, and future works

Hand-tracking systems have been widely studied and sometimes commercialized, but each with its own limitations in capturing the full dexterity of the human hand. In this work, we presented the contributions to this field of research, focusing on accurate estimation of the hand kinematic parameters and the hand motions, and also on robust reconstruction of various and even complex hand poses. We developed a wearable glove system, whose accuracy and robustness were improved by utilizing (1) a FK model tailored to the user's specific anatomical structure, (2) diverse and unrestricted hand motion data acquired through a custom-designed motion capture apparatus and postprocessing, and (3) a compact stretchable sensing mechanism that simultaneously measures the kinematic parameters and the joint motions. Through various quantitative evaluations (i.e., the estimations of the bone length, the joint angle, and the fingertip position), qualitative evaluations (i.e., the reconstructed hand poses and the Kapandji test), and practical applications (i.e., reconstruction of dexterous hand motions in virtual and real-world situations), we proved the distinguished performance of our hand tracking technology. Based on the contributions presented above and other criteria, we present a comparison of our technology to previous works and commercial products in Supplementary Table 5.

Our technology also has a few limitations. Compared to commercial products, there is room for further improvements regarding the processing speed, the convenient user interface, and machine

**Table 3 | Mean absolute errors of fingertip position estimation in *X*, *Y*, and *Z* directions and the 3D space**

| Subject | Mean absolute estimation error (mm) | | | | |
|---|---|---|---|---|---|
| | **X** | **Y** | **Z** | **3d** | **3d (Normalized)** |
| Small hand | 1.97 | 0.51 | 1.99 | 3.24 | 16.2% |
| Large hand | 2.15 | 0.75 | 2.81 | 4.02 | 16.1% |

The normalized 3D errors were calculated by dividing the 3D error with the side length of the workspace for each subject. Source data are provided in the Source Data file.

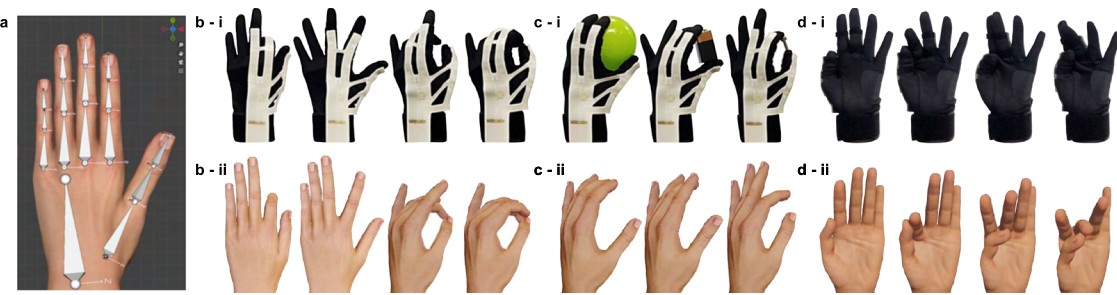

**Fig. 6 | Real-time hand pose reconstruction. a** Armature and mesh of the virtual hand created in an open-source 3D computer graphics software tool (Blender). **b (i)** Image of the wearer's real hand during various free hand motions and **(ii)** the corresponding real-time reconstruction of the hand poses in the virtual environment. **c (i)** Image of the wearer's real hand during three different grasping poses **(ii)** corresponding real-time reconstruction of the hand pose. **d (i)** Image of the wearer's real hand during modified Kapandji test **(ii)** corresponding real-time reconstruction of the hand pose.

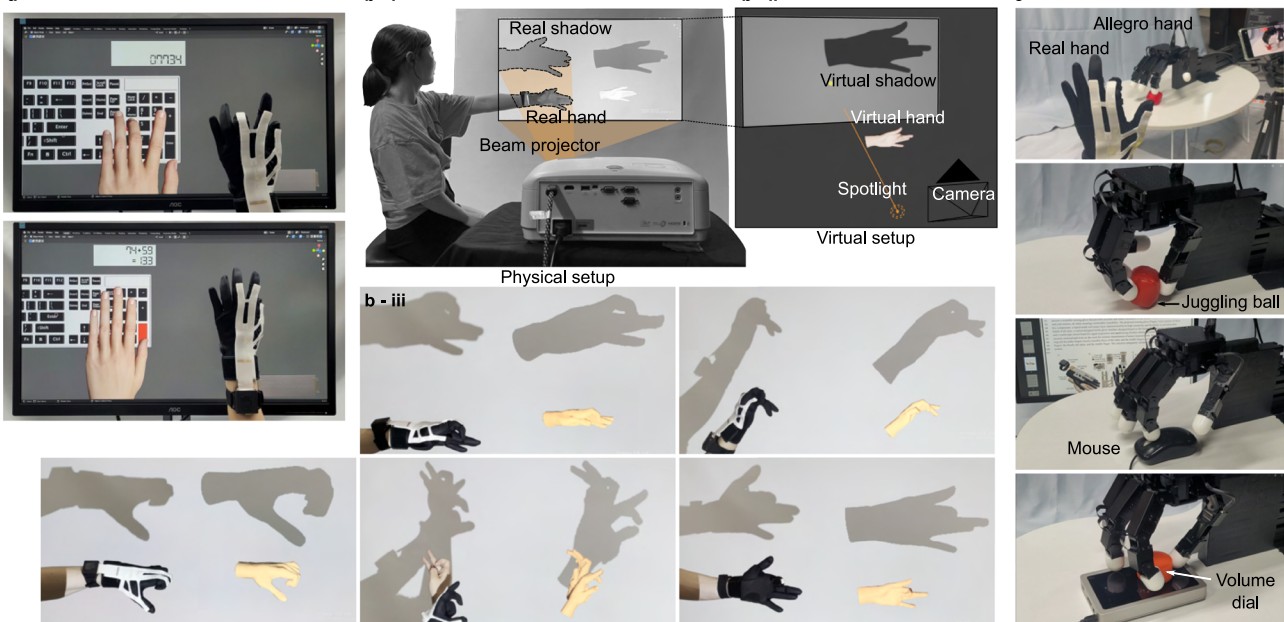

**Fig. 7 | Applications of the hand tracking system. a** Application of virtual number pad typing of five-digit number (top) and arithmetic operation (bottom). **b** Real world environment **(i)** and twin virtual environment (Blender) **(ii)** for real-time virtual shadowgraphy. **(iii)** Examples of virtual shadowgraphy showing the real hand and the shadow (left) and the virtual hand and shadow (right). **c** Teleoperation of an anthropomorphic robot hand (Allegro hand, Wonik Robotics) for three dexterous manipulations tasks: repositioning and grasping a ball, scrolling a mouse, and rotating a volume knob.

washability. In addition, we only demonstrated the sensing performance of three fingers since the ring and the little fingers have the same kinematic structure as the index and the middle fingers. While the same sensing mechanism and post-processing method can be extended to the tracking of the ring and the little fingers, this would have to overcome the increased processing cost and also require a more extensive training dataset. As was mentioned in the subsection Real-time refinement of bone lengths using signal distributions, the likelihood model for the sensor signals should be verified and adjusted based on a larger dataset. Moreover, the joint angle estimation accuracy may be further improved by adopting a nonlinear sensor model. Finally, our wearable glove can also benefit from implementing haptic feedback to provide a more immersive and intuitive control interface for robot teleoperation.

## Methods
### Design and positioning of eGaIn traces for measurement of joint motion
The nine eGaIn traces depicted in Fig. 1c were positioned so that the serpentine sensing patterns were positioned above the finger joints or in between the index and the middle finger and the thumb to accommodate hands of all sizes. Such design of the traces was intended to measure the following 10 DoFs joint motions: flexion of the proximal interphalangeal (PIP) and MCP joints of the index and middle fingers; flexion of the IP, carpometacarpal (CMC), and MCP joints of the thumb; and abduction of the MCP joint of the index and the middle fingers and the CMC joint of the thumb[61].

### Circuit board watch for wireless transmission of sensor measurements
As shown in Fig. 1a, a flat flexible cable (FFC) connects the nine eGaIn traces to a custom-designed and assembled printed circuit board (PCB). The PCB consists of programmable current sources (LT3092, Analog Devices) that feed the constant current into the sensors, the signals of which are read by analog-to-digital converters (ADS1115, Texas Instruments) and collected by the microcontroller unit (STM32F103C8T6, STMicroelectronics). The collected signals are transmitted wirelessly through a Bluetooth module (HC-06, Guangzhou HC Information Technology) to a host computer. Sensors are supplied with a constant current of 8 mA and sampled at 60 Hz. The

device was enclosed in a 3D-printed case (Onyx, Markforged) with straps attached for wearing on the wrist.

## Fabrication of the soft sensing layer

As depicted in Supplementary Fig. 1a, an Aluminum plate was coated with an elastomer layer (Dragon Skin 30, Smooth-On) with a thickness of 1.5 mm using a film applicator (4340, Elcometer), and cured in an oven at 60 °C for 15 min. This elastomer layer was then made into a mold by cutting the mold pattern using a laser cutter and removing the unnecessary parts, as shown in Supplementary Fig. 1b. Then, another layer of elastomer (Ecoflex 00-30, Smooth-On), mixed with a white pigment, was coated on top of the mold with a thickness of 1 mm above the mold surface and cured in the oven at 60 °C for 30 min. A 30 mm-long FFC was attached to the surface using a silicone adhesive (Sil-Poxy, Smooth-On), and the eGaln traces were automatically printed on the surface of the Ecoflex layer using a motorized X-Y stage (Shotmaster 300ΩX, Musashi), a pneumatic dispensing system (SuperΣ CMIII V2, Musashi), and a laser distance sensor (LK-G32, Keyence)[62]. The eGaln traces were directly printed from the silicone surface onto the electrodes of the FFC (Supplementary Fig. 1c)[46]. The eGaln traces were covered with a layer of transparent Ecoflex 00-30, 0.8 mm thick, and cured. Then, the outline of the sensing layer was cut using the laser cutter, as shown in Supplementary Fig. 1d. Finally, patches of hook-and-loop were bonded to the anchoring points on the back side of the sensing layer using an instant adhesive (Loctite), shown in Supplementary Fig. 1e. In designing the dimensions of the finger joint sensors, shown in Supplementary Fig. 5, we referred to the 1st percentile female index finger length as the lower limit[63]. This was to cover as large a range of hand sizes as possible so that the sensor would be subject to pre-stretch when worn by the wearer.

## Fabric selection and patterning in custom textile glove interface

An elastic fabric exhibiting small mass, high elongation, and recovery rates in both warp and weft direction were positioned from the knuckles to the tip of the fingers and also between the index finger and the thumb to allow comfortable joint articulations[64]. This fabric allowed the glove to be tightly fit over the hand even during dynamic motions while showing minimal residual elongation. The palm side of the hand was designed with an elastic mesh fabric with high air-permeability to ensure the breathability over prolonged use. We selected a combination of inextensible woven fabric and hook-and-loop straps to cover the majority of the back of the hand. The unique pattern shown in Fig. 1d was designed to allow an easy mode of wearing and removing the glove as well as to constrict the back of the hand so that all deformations of the glove are made only by the finger joint articulation. Rings of hook-and-loop straps were fastened onto the middle of the finger bone to provide firm anchor points that prevent slipping, rotation, and lifting of the sensor layer during dynamic hand motions. These rings were separate from the body of the glove so that their positions could be adjusted according to different finger and bone lengths as well as different circumferences. Our selection of fabrics as well as the testing categories for determining the fabric properties and the results are summarized in Supplementary Tables 1 and 2. Based on the average hand sizes of Korean men and women[65], we fabricated three glove sizes (S, M, L) to cover all hand sizes for male and female wearers.

## Variables, zero pose, and local frames in FK hand model for various hand anatomy

In Fig. 2b, the lengths of the distal, the middle, and the proximal phalanges (DP, MP, PP) and the MB of the thumb (T), the index (I), and middle (M) fingers are indicated. The IP and the MCP joints of the thumb and the PIP joint of the index and the middle fingers were modeled to have one-DoF flexion motions while the CMC joint of the thumb and the MCP joint of the index and the middle fingers have two DoFs including abduction[11,66,67]. We assumed there was no rolling

motion of the index and the middle fingers. In the zero pose, the index and the middle fingers were fully extended and aligned and the thumb was straightened with its CMC joint fully adducted. At this state, all fingers lie on the same plane of the back of the hand. The local frame of each bone in these two fingers was defined by its $x$-axis aligned with the bone lengths and the $z$-axis aligned with the axis of flexion. The base frame was attached to the back of the hand. This arrangement of the local frames was a general model that can be used to represent all hands of different anatomy. On the other hand, the joint axes of the thumb were not aligned with the other fingers and this misalignment varies between individuals[66]. In our model, we specified this variation by parameterizing the abduction offset ($\alpha_{abd}$) of the MB frame ($R_{meta}$) from the base frame ($R_{base}$) and the roll offset ($\gamma_{roll}$) of the proximal phalange ($R_{prox}$) from the MB ($R_{meta}$).

## Method for identifying hand anatomy (*True offsets*)

To identify the *true offsets* from the raw bone rotations obtained through the motion capture, we first assumed arbitrary offsets, $\alpha_{abd}$ and $\gamma_{roll}$, for the thumb MB ($\mathbf{R}_{meta}(\theta_{CMC,flex}=0)$) and the thumb proximal phalange ($\mathbf{R}_{prox}(\theta_{MCP,flex}=0)$). Then, the rotation of each bone during its flexion motion ($\mathbf{R}_{meta}(\theta_{CMC,flex})$, $\mathbf{R}_{prox}(\theta_{MCP,flex})$) can be expressed in the motion-capture global frame ($\mathbf{R}_s$) as a function of the flexion angle ($\theta_{CMC,flex}$ or $\theta_{MCP,flex}$) and the *true offsets* ($\alpha^*_{abd}$, $\gamma^*_{roll}$) as:

$$\begin{aligned}
\mathbf{R}_{meta}(\theta_{CMC,flex}) &= e^{[\boldsymbol{\omega}_{CMC,flex}]\theta_{CMC,flex}}\mathbf{R}_{meta}(0)\\
&= e^{[\boldsymbol{\omega}_{CMC,flex}]\theta_{CMC,flex}}\mathbf{R}_{base}\mathbf{R}_Y(\alpha_{abd})\\
&= \mathbf{R}_Y(\alpha^*_{abd})\mathbf{R}_Z(\theta_{CMC,flex})\mathbf{R}_Y(\alpha^*_{abd})^T\mathbf{R}_Y(\alpha_{abd})\\
&= \mathbf{R}_Y(\alpha^*_{abd})\mathbf{R}_Z(\theta_{CMC,flex})\mathbf{R}_Y(\alpha_{abd}-\alpha^*_{abd}) \quad (2)
\end{aligned}$$

$$= \mathbf{R}_Y(\theta_1)\mathbf{R}_Z(\theta_2)\mathbf{R}_X(\theta_3) \quad (3)$$

$$\begin{aligned}
\mathbf{R}_{prox}(\theta_{MCP,flex}) &= e^{[\boldsymbol{\omega}_{MCP,flex}]\theta_{MCP,flex}}\mathbf{R}_{prox}(0)\\
&= e^{[\boldsymbol{\omega}_{MCP,flex}]\theta_{MCP,flex}}\mathbf{R}_Y(\alpha^*_{abd})\mathbf{R}_X(\gamma_{roll})\\
&= [\mathbf{R}_Y(\alpha^*_{abd})\mathbf{R}_X(\gamma^*_{roll})]\mathbf{R}_Z(\theta_{MCP,flex})[\mathbf{R}_Y(\alpha^*_{abd})\mathbf{R}_X(\gamma^*_{roll})]^T\\
&\quad \mathbf{R}_Y(\alpha^*_{abd})\mathbf{R}_X(\gamma_{roll})\\
&= \mathbf{R}_Y(\alpha^*_{abd})\mathbf{R}_X(\gamma^*_{roll})\mathbf{R}_Z(\theta_{MCP,flex})\mathbf{R}_X(\gamma_{roll}-\gamma^*_{roll}) \quad (4)
\end{aligned}$$

$$= \mathbf{R}_Y(\alpha^*_{abd})\mathbf{R}_X(\theta_1)\mathbf{R}_Z(\theta_2)\mathbf{R}_Y(\theta_3) \quad (5)$$

while the base frame is aligned with the global frame ($\mathbf{R}_{base} \equiv \mathbf{R}_s = \mathbf{I}$). Here, $\omega_{CMC,flex}$ and $\omega_{MCP,flex}$ represent the rotation axes of the thumb MCP and IP joints, respectively. If the arbitrary offsets equal the *true offsets* (i.e., $\alpha_{abd} = \alpha^*_{abd}$ and $\gamma_{roll} = \gamma^*_{roll}$), the last terms in Eqs. 2 and 4 become identity. The rotation matrices $\mathbf{R}_{meta}$ and $\mathbf{R}_{prox}$ can be obtained through the motion-capture system and expressed as YZX and XZY Euler angles, as shown in Eqs. 3 and 5. Here, the elements of the Euler angles ($\theta_1$, $\theta_2$, $\theta_3$) represent the offset, the flexion angle, and zero, respectively. By sweeping the offset values, we can identify the *true offsets* as the values that minimize $\theta_3$.

## Test procedure and characterization method for initial bone length estimation

The experimental procedures are approved by the Seoul National University Institutional Review Board (IRB No. 2207/004-005, approved 20/07/2022). Thirteen subjects (nine males, four females, ages 23–31, with informed consent) were instructed to put on and remove the sensing layer five times, during which the sensor signals were alternately collected when the glove was worn (hand at zero pose) and removed (sensor at rest) (Supplementary Fig. 2). The bone

lengths of each subject were measured by a commercial image analysis software package (ProAnalyst, Xcitex) (Supplementary Fig. 3). As expected, increased sensor signals were observed in larger hand sizes and bone lengths. We then used the Gaussian process regression which is a multivariate nonparametric regression method to predict the bone lengths from the sensor signals. The mean function of the regression model was assumed to be quadratic in consideration of the hyperlinear nature of the microfluidic sensors and the squared exponential kernel function was used[47,68]. To evaluate the prediction, leave-one-out cross-validation was conducted for each subject.

### Real-time refinement of bone lengths using signal distributions

Once the wearer starts moving the fingers, the change in the sensor signals form a unique distribution depending on the hand size, which can be used to further refine the initial estimations. From Bayes rule[58], the bone lengths ($\mathbf{x}$) estimation can be updated given the subsequent sensor signals ($\mathbf{y}$) as

$$p(\mathbf{x}|\mathbf{y}) \sim p(\mathbf{x})\,p(\mathbf{y}|\mathbf{x}) \qquad (6)$$

assuming that the a priori distribution $p(\mathbf{x})$ is obtained from the initial estimation and the likelihood distribution of the sensor signal, $p(\mathbf{y}|\mathbf{x})$, was characterized as a function of the bone lengths[69]. To characterize the likelihood, we conducted additional experiments with the thirteen subjects (nine males, four females, ages 23–31, with informed consent) and collected sensor signals during dynamic hand motions. To simplify the problem, the hand motions were constrained to the poses illustrated in Fig. 3a to ensure that the likelihood in Eq. 6 can be fitted as Gaussian. For each subject, the mean ($\boldsymbol{\mu} \in \mathrm{R}^9$) and variance ($\boldsymbol{\sigma_i} \in \mathrm{R}^9$) of the collected sensor signals were calculated and then linearly fitted to the subject's bone lengths ($\mathbf{x} \in \mathrm{R}^{10}$) as

$$\boldsymbol{\mu} \cong \mathbf{M}\mathbf{x} = \hat{\boldsymbol{\mu}} \qquad (7)$$

$$\mathrm{Diag}(\boldsymbol{\sigma_i}) \cong \mathbf{V}\mathbf{x} = \hat{\boldsymbol{\Sigma}} \qquad (8)$$

An example of the fitted result is presented in Fig. 3b, showing the histogram of one sensor signal with its Gaussian approximation, $\mathrm{N}(\boldsymbol{\mu},\mathrm{Diag}(\boldsymbol{\sigma_i}))$, defined by the true mean and variance of the sensor signals, and the estimated distribution, $\mathrm{N}(\hat{\boldsymbol{\mu}},\hat{\boldsymbol{\sigma}})$, defined by the estimated mean and covariance calculated from the subject's bone lengths. From the Gaussian assumption, the *a posteriori* distribution can be obtained in an analytic form as

$$\hat{\mathbf{x}}_k = \hat{\mathbf{x}}_{k-1} + \mathbf{P}_{k-1}\mathbf{M}^{\mathrm{T}}\left(\mathbf{M}\mathbf{P}_{k-1}\mathbf{M}^{\mathrm{T}} + \hat{\boldsymbol{\Sigma}}\right)^{-1}\left(\mathbf{y}_k - \mathbf{M}\hat{\mathbf{x}}_{k-1}\right) \qquad (9)$$

$$\mathbf{P}_k = \left(\mathbf{I} - \mathbf{P}_{k-1}\mathbf{M}^{\mathrm{T}}\left(\mathbf{M}\mathbf{P}_{k-1}\mathbf{M}^{\mathrm{T}} + \hat{\boldsymbol{\Sigma}}\right)^{-1}\mathbf{M}\right)\mathbf{P}_{k-1} \qquad (10)$$

Here, the equation is expressed in recursive form since the estimation is continuously updated in real time as the sensor signals arrive sequentially ($\ldots, \mathbf{y}_{k-1}, \mathbf{y}_k$). Result for three of the subjects (two male, one female, ages 23–27) have been depicted in Fig. 3c, d.

### Implementation of the linear system model in joint angle estimation

The linear system model in Eq. 1 was based on two conditions. First, bias terms of the joint angles and the sensor signals were removed from the state ($\mathbf{x}_k$) and the observation ($\mathbf{y}_k$). The joint angle bias refers to the non-zero thumb CMC abduction ($\alpha_{\mathrm{abd}}$) at the zero pose, and the sensor bias is the initial sensor signal that was used to estimate the bone lengths. By removing the initial sensor signal, the joint angle estimation was separated from the bone length estimation. Second, we

assumed a linear sensor model. This assumption is grounded on the observation that the relationships between the sensor signals and their corresponding joint angles are approximately linear within the range of motion of the joint (Supplementary Fig. 4). However, since liquid-metal strain sensors display a hyperlinear behavior in large deformations, the gradient of the linear relationship must be modified to compensate the different levels of pre-stretching by different wearers. In this study, we categorized the subjects into small and large hands, and the system matrices ($A$, $Q$, $H$, $R$) of each group were calibrated by fitting the sensor data and the ground-truth calibration dataset to Eq. 1 by the least-square method.

The sensor signals were processed and the joint angles were estimated using a Kalman filter. We evaluated the estimations using the test data (~65 s, Fig. 4b). Two subjects were tested in the *self-calibration* group (one male, one female, 26–27, with informed consent), and two subjects were tested in the *non-calibration* group (one male, one female, ages 25–29, with informed consent).

### Test setup and procedures for prediction of fingertip position

The tests were conducted for two subjects (one male, one female, 26–27, with informed consent) with different hand sizes. Each subject was provided trajectories of a square with side lengths of 25 mm and 20 mm and instructed to follow the traces with their index finger and thumb, as shown in Fig. 5a. The sizes of the trajectories were selected based on the workspace of each subject's fingers. In addition, for uniform sampling of the test data for unbiased evaluation, the subjects were guided by periodic buzzer sound (every three seconds) between each stroke.

### Setup of twin (real and virtual) environments for virtual shadowgraphy

As shown in Fig. 7b, we first designed a 3D virtual environment with a strong light casted against a hand model to project its shadow onto the vertical plane at the back. A twin environment was set up in the real world with a projector and a screen so that the wearer could simultaneously cast the hand shadow onto the screen where the virtual hand and its shadow were also displayed for comparison. While the postures of the index and the middle fingers and the thumb were updated in real-time based on the sensor glove's estimations, the postures of the ring and the little fingers were fixed to the desired configuration since their precise postures were also required in this application. For the case of the rabbit, an additional virtual right hand was added, fixed in a similar way to the ring and the little fingers.

### Reporting summary

Further information on research design is available in the Nature Portfolio Reporting Summary linked to this article.

## Data availability

The datasets generated in this study have been provided in the Source Data file and have also been deposited in the Figshare database under accession code 10.6084/m9.figshare.25734636[70]. Source data are provided with this paper.

## Code availability

The codes generated in this study have been deposited in the Figshare database under accession code 10.6084/m9.figshare.25734636[70].

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

## Acknowledgements

This work was supported by the National Research Foundation of Korea (Grant No. RS-2023-00208052) funded by the Korean Government (MSIT), Technology Innovation Program (Grant No. 20008912) funded by the Ministry of Trade, Industry & Energy (MOTIE, Korea), and Institute of Information & communications Technology Planning & Evaluation (IITP) (Grant No. 2021-0-00896) grant funded by the Korea government (MSIT).

## Author contributions

M.P. and Y.L.P. conceived the idea. M.P. and T.P. designed and fabricated the sensing hardware. S.J.Y. designed and fabricated the wireless PCB. S.P. designed and fabricated the textile glove interface. M.P., T.P., and S.J.Y designed the virtual environment for experiments and applications. M.P. and T.P. conducted the experiments and analyzed the data. M.P. and T.P. drafted the manuscript, and all the other authors contributed to writing and revising the manuscript. S.H.K. supervised the development of the glove interface. Y.L.P. acquired funding, provided resources, and supervised the overall research.

## Competing interests

A provisional patent application (Serial No.: 10-2024-0075114) related to part of this work was filed on June 10, 2024 in Korea (ROK). A provisional patent application (Serial No.: 10-2024-0075114, Inventors: Yong Lae Park, Myungsun Park, Taejun Park, Sohee John Yoon) related to part of this work was filed on June 10, 2024 in Korea (ROK). The remaining authors declare no competing interests.
