## [Peer Review File · Nature Communications]

Stretchable glove for accurate and robust hand pose reconstruction based on comprehensive motion dataREVIEWER COMMENTS

Reviewer #1 (Remarks to the Author):

It seems that the pinky finger is not sensed by the glove. Therefore, most of the hand gesture is only limited to 4 fingers.

I am not too enthusiastic about the ability to sense the length of the finger as it is mainly able to detect based on relative size from the original glove. Therefore, if the hand is smaller than the glove it is not able to sense it. I do not see any application for this and is not a novel idea.

Further, data-glove has been a common device and a company called StretchSense have a much more complete device. Hence, I do not see any novelty for this to be published in this esteem journal. Unfortunately, I have to recommend that this manuscript to be declined for publication.

Reviewer #2 (Remarks to the Author):

The paper is well-written, with clear illustrations of the work. Based on the results presented, I have no objections to the soundness of this work. Considering that the development of data gloves spans half a century, leading to numerous proposed designs with new ones continually emerging, I am concerned that the authors may not have delved deep enough into the subject.

Stretchable materials commonly face a degradation problem, as their properties tend to change over time with repetitive usage, leading to severe reliability and robustness issues. However, this aspect is not thoroughly evaluated in the paper. I recommend that the authors refer to the following paper for an assessment of sensor response after numerous cycles:

Sundaram et al., "Learning the signatures of the human grasp using a scalable tactile glove." *Nature* (2019)

To calibrate the device and assess the estimation of joint angles/bone lengths, the authors have provided a list of hand gestures for reference. However, the rationale or principle behind the selection of these gestures is not sufficiently discussed. I would suggest that the authors consider adopting more standard metrics for hand gesture evaluation, such as the Kapandji score.

From the perspective of glove design, the proposed work appears to be essentially a layer of stretchable material augmented on top of an ordinary glove. To enhance the paper, the authors could consider focusing either on the material aspect, especially if it is a novel material, and conduct more evaluations on the material itself, akin to material science papers, or on the application side and provide a more in-depth analysis. Relevant examples of in-depth analyses can be found in the following papers:

Zhu et al., "Haptic-feedback smart glove as a creative human-machine interface (HMI) for virtual/augmented reality applications." *Science Advances* (2020)

Feng et al., "AI enabled sign language recognition and VR space bidirectional communication using triboelectric smart glove." *Nature Communications* (2021)

From the application perspective, the two applications presented in the paper are rather simple and lack significant meanings, more or less like toy examples. I would encourage the authors to explore more impactful applications by referring to the following work for inspiration:

For training AI: Liu et al., "A Reconfigurable Data Glove for Reconstructing Physical and Virtual Grasps." *Engineering* (2023).

For various downstream applications: Lee et al., "Visual-inertial hand motion tracking with robustness against occlusion, interference, and contact." Science Robotics (2021)

The use of FK to regulate hand pose sensing during hand gesture reconstruction is already a popular practice:

Liu et al., "A glove-based system for studying hand-object manipulation via joint pose and force sensing." IROS (2017)

Liu et al., "A new IMMU-based data glove for hand motion capture with optimized sensor layout." International Journal of Intelligent Robotics and Applications (2019)

Reviewer #3 (Remarks to the Author):

---WORK SUMMARY---

This work presents a novel data glove that allows both anthropometric hand measurement and joint kinematics recording of thumb, index and middle fingers. The design and technical validation of the glove are described. This work is innovative and may be valuable for researchers devoted to hand kinematics, wearable systems or robotics.

---REVIEW SUMMARY---

This study addresses an important issue present when recording hand kinematics. A common practice when analysing hand posture is using the same kinematic model for all data collected. This approach can be useful in applications such as ergonomics or hand functional evaluation, but when the fingertip position is crucial, kinematic hand models should ideally include user hand anthropometric parameters. While these data can be obtained using optical motion capture systems, their use is sometimes cumbersome, with a very time-consuming setup and data processing (I suffered it!). Furthermore, in recordings involving product manipulation, marker occlusions may appear in optical systems. For these purposes, data gloves are the most appropriate systems, but they commonly provide data only of hand joints, or entire hand posture in space. This work presents a novel data glove that allows recording hand kinematics along with hand anthropometric measurements, being of significance for the researchers devoted to hand kinematics, among other fields. The main content of the work is relevant and well structured. The methodology is well described in detail, and does not have any flaw. Minor revisions have been proposed, but none of them questioning experimental phase or validation performed. The use of English is correct.

---DETAILED REVIEW---

-TITLE-

The title clearly reflects the content of the work.

-ABSTRACT-

The abstract should somehow mention that the glove only measures thumb, index and middle fingers.

11: The word "dynamic" is a more general term that commonly involves also forces, mass, etc. It is used throughout the whole manuscript. In this specific case, maybe better just say "estimates finger joint motions" or "estimates finger joint kinematics". Even though, this sentence seems quite similar to the first sentence of the abstract and could be both merged.

14-16: The sentence is difficult to follow, I would suggest proposing an alternative phrasing.

The abstract could benefit of a sentence mentioning the potential applications of the glove or the fields of study that would be interested in this glove.

-INTRODUCTION-

The introduction presents appropriately the state of the art and evidences the necessity to develop the glove.

25: I miss some references of the mentioned applications.

28: I suggest substituting the word "rudimentary" for another without such level of negative connotations.

The introduction should also mention that the presented glove only measures thumb, index and middle fingers.

-RESULTS-

97: The kinematic structures of the ring and little fingers resemble to those of index and middle fingers, but this does not mean that they could be estimated from index and middle fingers motion. In fact, they present different motion patterns when analysing hand kinematics during varied tasks. They only can be assumed similar in power grasps (cylindrical, lumbrical, spherical, etc), but not in precision grasps, where the main role is assumed by thumb, index and maybe middle finger. Therefore, I suggest specifying "since the kinematic structures of the ring...". Even though, the fact that the behaviour of these two fingers is not studied should be mentioned as a limitation of the work and maybe proposed as a future work.

103: Joint abbreviations used in figure 1 are missing in the caption or main text (specify CMC for carpometacarpal, MCP for metacarpophalangeal, etc.).

103: Does the index-middle abduction sensor hinder motion? Can the index finger fully cover its MCP flexion range while middle MCP remains at neutral position? From the supplementary material videos it cannot be entirely appreciated. It is just a personal concern.

169: The bone abbreviations are missing (DP, MP, PP and MB).

206-211: Symbols "?" in equations.

223: I would also comment here the assumption considered to represent ring and little fingers pose. Was it also assumed as two thirds of the middle finger joints?

246: Add to the caption something like: "Bone abbreviations as in Figure 2".

261-268-269-274-275: Symbols "?" in equations.

298-299: Maybe "covering the full range of motion of hand joints".

379-404: The videos are well prepared and clearly illustrate the applications tested, good job.

-METHODS-

The methods section presents all the analyses and procedures in detail and no important information is apparently missing. It is supported by the supplementary material, which provides details about the characteristics of the materials used, data collected during the several validation studies or key frames from the test applications videos.

In this section start to appear the abbreviations missing in the results section. Some of them should be defined there, in results section, to ease the manuscript readability.

-ADDITIONAL SUB-SECTION-

This manuscript could highlight the main strengths of the work presented by adding a sub-section remarking the main advantages of the glove designed regarding other recording methods, its applicability in several research fields and its limitations. Furthermore, this section could outline the future works, such as different angle estimation approaches (as mentioned in line 281) or studying also ring and little fingers. Maybe this could be a final sub-section in the results section. Something like "novelty, applicability and limitations". It is just a suggestion, not a requirement.

Reviewer #4 (Remarks to the Author):

The authors propose an interesting system for finger tracking, but some of the declared advantages are not demonstrated, and the declared advantages with respect to other technologies are not quantified.

Nevertheless, with major revisions this work can achieve some interest for the readers.

I. Introduction

Q1: "rudimentary results" is a too generic sentence. Please provide justification/details.

Q2: Similarly, it is convenient to furnish some numbers/order of magnitude regarding the affirmed "precise localization of the fingertips and detailed posture of the hand".

Q3: "strain/bending sensors, encoders, or inertial measurement units (IMUs) ... vision or magnetic sensors" are the mostly adopted but not exclusive, and other approaches have been reported.

Q4: "these methods of measurements require highly controlled environmental setups" is true for vision-based methods only. It is not true, for instance with the adoption of "strain/bending sensors"

Q5: "limiting the wearer's free hand motions" and "this inevitably results in bulky and restrictive systems with reduced wearability" are not issues with not-mentioned technologies, such as bone conduction-based system (see: Saggio, G., Santoro, A. S., Errico, V., Caon, M., Leoni, A., Ferri, G., & Stornelli, V. (2021). A novel actuating-sensing bone conduction-based system for active hand pose sensing and material densities evaluation through hand touch. IEEE Transactions on Instrumentation and Measurement, 70, 1-7.).

Q7: how overcoming "limiting the wearer's free hand motions" is gained with your solutions? It is not mentioned in the following

Q8: "we propose a wearable glove ... with high accuracy"

Q9: "our system not only demonstrates a compact design" with respect to what other systems?

II. Results

Q10: "accurate and robust estimation" are claimed, but not mentioned in terms of what: reproducibility? Replicability? Sensitivity? Precision? ..

Q11: "a liquid-metal soft sensor layer characterized by its high sensitivity and flexibility": what does it mean "high sensitivity"? "High" is an adjective not a measurement and "flexibility" is not defined with a number.

Q12: "to accommodate hands of all sizes": hands differ not only in length but is diameter too. How this issue is considered?

Q13: "Since the kinematics of the ring and the pinky fingers closely resemble those of the index and the middle fingers": in attached videos, the pinky finger does not move "closely resemble", please clarify. Moreover your sentence is not always true. In addition, what does "closely resemble" mean? In terms of what?

Q14: "large deformations while exerting minimal force": how "large" and how "minimal"?

Q15: "The mechanical structure of the sensing layer was designed to achieve the increased resolution, resulting in a more accurate measurement of the bone lengths and the joint angle": "more accurate" with respect to what?

Q16: ". By securing the layer onto the middle of each finger bone, each sensor was positioned

directly over each finger joint”: did you consider possible misalignment issues? (see: Saggio, G. (2014). A novel array of flex sensors for a goniometric glove. *Sensors and Actuators A: Physical*, 205, 119-125.)

Q17: You report “estimation errors” (Table 1), but you claim “accuracy” that must be addressed in terms of repeatability, reproducibility and reliability too, please add these missing parts, referring to a number of published papers addressing these concerns since Wise (Wise, S., Gardner, W., Sabelman, E., Valainis, E., Wong, Y., Glass, K., ... & Rosen, J. M. (1990). Evaluation of a fiber optic glove for se-automated goniometric measurements. *J Rehabil Res Dev*, 27(4).)

Reviewer #1

- *It seems that the pinky finger is not sensed by the glove. Therefore, most of the hand gesture is only limited to 4 fingers.*

We appreciate your comment. We realized the specific fingers that are sensed by the sensing glove may not have been clearly presented in the original manuscript. The proposed sensing glove presents accurate tracking of the motion of the thumb, index and middle fingers. Since the joints of the ring and little fingers follow the same kinematic structure (Fig. 2 (b)) as that of the index and middle finger, we demonstrated the accuracy and reliability of our hand motion sensing mechanism using only the index, middle fingers and thumb to prove the concept of our sensing glove. Please note all quantitative evaluations made in this research were restricted to only these three fingers. In reconstructing the hand poses using the virtual hand model, the joint angles of the ring and little fingers were assumed as the angles of the middle finger joints multiplied by $\frac{2}{3}$ and $\frac{1}{3}$ respectively. This was done only to visualize more natural hand poses and were not a result of direct estimation of the corresponding finger joints so were not included in the evaluation of the sensing glove's performance. The system can be easily extended to measure the lengths and motions of the ring and little fingers using additional sensors with the same sensing mechanism and post-processing method.

We understand such characteristics of the glove may not have been clear in the original manuscript, so we have edited the Abstract, Introduction, and the Results sections to clarify this, as reproduced below:

Abstract: "The soft sensing glove is designed to easily stretch and to be one-size-fits-all, measuring both the size of the hand and estimating finger joint motions of the thumb, index, and middle fingers."

Introduction: "The glove measures the lengths and the motions of the thumb, the index, and the middle fingers. However, the same sensing mechanism may be extended to measure the rest of all five fingers that share the same kinematic structure."

Results: "Since the kinematic structure of the ring and the little finger joints resemble those of the index and the middle fingers, we only test the sensing performance for estimating the lengths and motions of the thumb, the index and the middle fingers to prove the concept of our study. This was considered to be adequate in validating the performance of the proposed system."

...

"These two fingers were not included in the evaluation of the performance of the glove system, which is a limitation of the glove in its current state since other variations of precision grasps may require independent motions of each finger for example. For a more comprehensive estimation of the full hand motion, the system can be easily extended using additional sensors and the same post-processing method, which we pose as future work."

- *I am not too enthusiastic about the ability to sense the length of the finger as it is mainly able to detect based on relative size from the original glove. Therefore, if the hand is smaller than the glove it is not able to sense it. I do not see any application for this and is not a novel idea.*

Thank you for the thoughtful comment. In designing the sensing layer, consisting of the liquid metal sensing channels, the dimension of finger joint sensors were designed so that when attached on the textile glove interface worn by the wearer, the sensors would always be pre-stretched. The dimensions of the sensor layer were selected to cover as large a range of adult hand sizes as possible, referring to anthropometric data of the human hand and using the 1st percentile female index finger length as the lower limit (Tilley, Alvin R. The measure of man and woman: human factors in design. John Wiley & Sons, 2001). However, since the glove was designed for adult hands, it would not be capable of sensing finger length of smaller hands such as those of children in its current state. For those cases, a smaller glove design is required. These information about the glove design were not provided in the previous manuscript, so we have added discussions to Methods - *Fabrication of the soft sensing layer* of the manuscript as reproduced below:

“In designing the dimensions of the finger joint sensors, shown in Figure S5 in *Supplementary Information*, we referred to the 1st percentile female index finger length as the lower limit [65]. This was to cover as large a range of hand sizes as possible, so that the sensor would be subject to pre-stretch when worn by the wearer.”

- *Further, data-glove has been a common device and a company called StretchSense have a much more complete device. Hence, I do not see any novelty for this to be published in this esteem journal. Unfortunately, I have to recommend that this manuscript to be declined for publication.*

We agree the wearable gloves from StretchSense presents a reliable and fast mode of sensing motions of the hand that can even be integrated with other commercial motion capture systems. However, we still believe our sensing glove presents novel aspects that differentiate it from other commercial gloves. First, very little is reported on the quantitative evaluation on the accuracy of stretchsense gloves. While case studies present Stretchsense gloves can drastically reduce the production time of animating hand motions, specific quantitative evaluation results of the accuracy or error were not provided. However, our research presents in-depth analysis of the accuracy of our proposed glove system in estimating the finger joint angles and fingertip position as well as qualitative evaluation of the reconstructed hand poses. We believe the experimental protocols with which we evaluated the sensor performance, using customized optical motion capture system and rigid body marker set ups also presents new contributions to evaluating performance of sensing gloves by enabling collection of a comprehensive data set of natural hand motions expressed through joint angles and fingertip position, which can further advance the related technologies.

Secondly, while StretchSense requires users to purchase different glove sizes depending on finger length and palm circumference, our proposed glove system can cover a wide range of hand sizes (as mentioned in the response to the previous comment) and also provides a means of estimating the wearer’s finger lengths, which can be used to reconstruct a more accurate virtual hand model or animation by precisely estimating the fingertip position. The integration of finger joint angle sensing and finger length estimations in a single sensing mechanism is presented as another novelty of our proposed glove system, allowing for accurate reconstruction of the wearer’s hand motions.

However, there are limitations to our sensing glove that are overcome by Stretchsense, such as its durability regarding machine washability, tracking and processing speed, and convenient user interface. We believed the previous manuscript could benefit from a more detailed comparison of our glove system to other commercial sensing gloves, and so an additional subsection was provided to analyzing the contributions and limitations of our sensing glove with respect to other previous works and commercial hand tracking gloves in the subsection *Contributions, Limitations, and Future works* and Table S5 in *Supplementary Information*:

“**Contributions, limitations, and future works**

Hand tracking systems have been widely studied and sometimes commercialized, but each with its own limitations in capturing the full dexterity of the human hand. In this work, we presented the contributions to this field of research, focusing on *accurate* estimation of the hand kinematic parameters and the hand motions, and also on *robust* reconstruction of various and even complex hand poses. We developed a wearable glove system, whose accuracy and robustness were improved by utilizing 1) a forward kinematic model tailored to the user’s specific anatomical structure, 2) diverse and unrestricted hand motion data acquired through a custom-designed motion capture apparatus and postprocessing, and 3) a compact stretchable sensing mechanism that simultaneously measures the kinematic parameters and the joint motions. Through various quantitative evaluations (i.e., the estimations of the bone length, the joint angle, and the fingertip position), qualitative evaluations (i.e. the reconstructed hand poses and the Kapandji test), and practical applications (i.e., reconstruction of dexterous hand motions in virtual and real-world situations), we proved the distinguished performance of our hand tracking technology. Based on the contributions presented above and other criteria, we present a comparison of our technology to previous works and commercial products in Table S5, *Supplementary Information*.

Our technology also has a few limitations. Compared to commercial products, there is room for further improvements regarding the processing speed, the convenient user interface, and machine washability. In addition, we only demonstrated

the sensing performance of three fingers since the ring and the little fingers have the same kinematic structure as the index and the middle fingers. While the same sensing mechanism and post processing method can be extended to the tracking of the ring and the little fingers, this would have to overcome the increased processing cost and also require a more extensive training dataset. As was mentioned in the subsection of *Real time refinement of bone lengths using signal distributions*, the likelihood model for the sensor signals should be verified and adjusted based on a larger dataset. Moreover, the joint angle estimation accuracy may be further improved by adopting a nonlinear sensor model. Finally, our wearable glove can also benefit from implementing haptic feedback to provide a more immersive and intuitive control interface for robot teleoperation.”

Furthermore, to highlight the exceptional accuracy of our system and also a more practical use of the sensing glove, we demonstrated an additional application: teleoperation of a dexterous robot hand for precise manipulation of various objects in practical situations where accurate control of the fingertips and hand posture is crucial. Description of the new application is reproduced below, and results of the application are shown in Supplementary Video 6.

“Application 3: Teleoperation of dexterous robotic hand

We also performed teleoperation of an anthropomorphic robotic hand (Allegro Hand, Wonik Robotics), for precise manipulation of various objects in practical situations (Figure 7(c), Supplementary Video 6). We carried out three different tasks that require dexterous and precise manipulation. The first was manipulation of a ball on a table, repositioning the ball to a desired location and grasping it. The next task is to control the robotic hand to manipulate a mouse wheel, to scroll through pages of a document. This task required accurate control of the fingertip position so that the finger only rotates the mouse wheel without pressing it and activating a different mouse function. Finally, a volume control knob of an audio interface was manipulated by the robotic hand, which required unique sequences of multiple finger motions to rotate the cylindrical knob in both directions. The results shown in Supplementary Video 6 demonstrate that our glove system can be used to take full advantage of the dexterity of the human hand in various practical scenarios.”

Reviewer #2

- *The paper is well-written, with clear illustrations of the work. Based on the results presented, I have no objections to the soundness of this work. Considering that the development of data gloves spans half a century, leading to numerous proposed designs with new ones continually emerging, I am concerned that the authors may not have delved deep enough into the subject.*

Stretchable materials commonly face a degradation problem, as their properties tend to change over time with repetitive usage, leading to severe reliability and robustness issues. However, this aspect is not thoroughly evaluated in the paper. I recommend that the authors refer to the following paper for an assessment of sensor response after numerous cycles:

Sundaram et al., "Learning the signatures of the human grasp using a scalable tactile glove." Nature (2019)

We thank you for the thoughtful comments. As the reviewer has mentioned, soft stretchable sensors often face issues regarding structural robustness and reliability, which can limit more practical applications or commercialization of the sensors. We also realized this could be a potential problem to our sensing glove and so have invested great effort in designing and also proving a reliable and robust sensing glove system.

Our sensing layer, composed of silicone and liquid-metal (LM) sensing channels, demonstrates enhanced sensitivity compared to other more common LM-based stretch sensors due to its multi-stiffness matrix structure. This LM soft sensor structure was adopted from our previous work (Myungsun Park, Taejun Park, and Yong-Lae Park. "Parametric analysis of multi-material soft sensor structures for enhanced strain sensitivity." *Extreme Mechanics Letters* 60 (2023): 101983), which presented a mechanism by which the sensitivity of single material LM sensors can be improved by utilizing a substrate structure composed of varying stiffness. In this previous work, the structural robustness of the improved sensor was tested by stretching and releasing the sensor 10,000 times while monitoring the sensor signal, and we confirmed that the sensor displayed reliable stability in repeated use. Since we believed this work sufficiently demonstrated the reliability and robustness of the multi-stiffness soft stretchable sensors, we did not include similar cyclic test results in the current manuscript. Instead, we presented other evaluations of repeatability and reliability of the soft, stretchable sensing layer. Since the same sensing glove can be worn and be used to estimate the finger lengths and hand motions of various hand sizes, we considered it necessary to test and prove the repeatability of the sensor measurements when worn on the wearer's hand. This ensures the reliability of the sensor in representing the user's hand size and hand motion without additional calibration processes. So the sensor signal was monitored while it was repeatedly worn and removed on various subjects with different hand sizes as discussed in Methods - *Testing procedure and characterization method for initial estimation of bone lengths using initial sensor signal* and Figure S2 in Supplementary Information, *Sensor signal measurements for different hand size*:

“Each subject was instructed to put on and remove the sensing layer five times, during which the sensor signals were alternately collected when the glove was worn (*hand at zero pose*) and removed (*sensor at rest*)”

“Sensor signals for each of the nine sensors (S1-S9) measured at the initial unstretched state of the sensor (white), when worn on a small hand (light), and when worn on a large hand (dark), showing error bars obtained over five trials.”

However, we agreed it could benefit the paper if we discussed these concerns regarding the reliability of soft stretchable sensors and also our approach in tackling this potential problem. Therefore, we have added the following details and edited the subsection *Soft and sensitive liquid metal-based sensor* as reproduced below:

“The mechanical structure of the sensing layer was designed to **increase the resolution of liquid-metal soft strain sensors**, resulting in accurate measurement of the bone lengths and the joint angle. One approach to increase the sensitivity of the liquid-metal strain sensors is to design the substrate with a combination of multiple elastomer strips with different

stiffnesses [48]. These sensors are also known for their reliability in prolonged and repeated loading cycles, which overcomes the structural limitation often faced by soft, stretchable sensors. We thus employed this approach, creating the stiffness variations by altering the thickness of the substrate with a single material, as shown in Figure 1(b). This method not only enhanced the structural integrity of the sensor but also improved the sensitivity (gauge factor: 3.4).”

- *To calibrate the device and assess the estimation of joint angles/bone lengths, the authors have provided a list of hand gestures for reference. However, the rationale or principle behind the selection of these gestures is not sufficiently discussed. I would suggest that the authors consider adopting more standard metrics for hand gesture evaluation, such as the Kapandji score.*

Thank you for the constructive feedback. We originally selected the specific set of hand poses to cover all range of motions of the thumb, index, and middle finger joints independently as well as a variety of their combinations. However, we agree that adopting a more standard metric for the evaluation could be a more effective method of demonstrating the performance of our sensing glove, so we adopted the Kapandji test in our evaluation. However, since our glove only directly measures and estimates the motion of the thumb, index and middle fingers, and all evaluations of the sensing glove were conducted on these fingers only, the Kapandji test was modified to test the reach of the thumb to various portions of the index and middle finger. We added descriptions and discussed the results in Section II. Results, *Real-time hand pose reconstruction* as reproduced below, and have also added the video of the test results in Supplementary Video 3.

“We also conducted a more standardized test to evaluate the accuracy of the glove system in reconstructing opposition of the thumb and its interaction with other fingers using the Kapandji score [19]. However, since the glove system only directly estimates the motions and the lengths of three fingers (thumb, index, and middle finger), and accurate reconstruction of the ring and little finger are not evaluated in this research, the test was modified so that the thumb makes contact with the DP, MP, and PP bones of the index and the middle fingers only. Therefore, the original scoring metric was not applied to this modified test. The results (Figure 6(d) and Supplementary Video 3) suggest the proposed glove is capable of sensing and reconstructing the opposition of the thumb as well as accurately reconstructing various forms of interactions (contacts) of the fingers.

”

- *From the perspective of glove design, the proposed work appears to be essentially a layer of stretchable material augmented on top of an ordinary glove. To enhance the paper, the authors could consider focusing either on the material aspect, especially if it is a novel material, and conduct more evaluations on the material itself, akin to material science papers, or on the application side and provide a more in-depth analysis. Relevant examples of in-depth analyses can be found in the following papers:*

Zhu et al., "Haptic-feedback smart glove as a creative human-machine interface (HMI) for virtual/augmented reality applications." Science Advances (2020)

Feng et al., "AI enabled sign language recognition and VR space bidirectional communication using triboelectric smart glove." Nature Communications (2021)

From the application perspective, the two applications presented in the paper are rather simple and lack significant meanings, more or less like toy examples. I would encourage the authors to explore more impactful applications by referring to the following work for inspiration:

For training AI: Liu et al., "A Reconfigurable Data Glove for Reconstructing Physical and Virtual Grasps." Engineering (2023).

For various downstream applications: Lee et al., "Visual-inertial hand motion tracking with robustness against occlusion, interference, and contact." Science Robotics (2021)

We initially presented the main contributions of this research as the integration of finger joint angle sensing and finger length estimations in a single sensor for accurate reconstruction of the hand motion, calibration and evaluation of the glove system using comprehensive and novel ground truth data, and a robust system capable of estimating arbitrary hand poses. However we have taken the reviewer's advice to further strengthen the contribution of this paper by demonstrating practical and in-depth analysis of the performance of the sensor through a new application.

An important aspect of our glove system that we wish to highlight is its robustness and accuracy in estimating arbitrary and complex hand motions to reflect the dexterity of the human hand. Therefore, we chose to demonstrate teleoperation of the Allegro Hand (Wonik Robotics), a dexterous robotic hand, for precise manipulation of various objects in practical situations. Wearable hand motion sensing gloves provide an intuitive and thereby effective method of controlling robotic hands by directly replicating the controller's hand motions with the robot. If the hand motions can be reconstructed with high accuracy, the robot hand may be controlled with high precision in performing delicate tasks. We intended to highlight such aspects of our glove system through this new application. We carried out three different tasks: 1) adjusting the position and grasping a ball on a table to demonstrate dexterous object manipulation, 2) scrolling a mouse wheel, which requires accurate control of the fingertip position so that the finger only rotates the mouse wheel without pressing it and activating a different mode, and 3) adjusting the volume dial of an audio interface, which requires unique set of finger motions to rotate the cylindrical knob in both directions. These tasks extend beyond the scope of simple pick and place demonstrations, that often rely more on accurate positioning of the hand rather than the precise motions of the fingers. Therefore, this application proves the glove system can accurately and robustly reconstruct dexterous hand motions, which is effective for teleoperating robotic hands in various practical situations.

Discussions of this new application was added to the subsection *Application 3: Teleoperation of dexterous robotic hand* as reproduced below, and the results are shown in Supplementary Video 6:

“Application 3: Teleoperation of dexterous robotic hand

We also performed teleoperation of an anthropomorphic robotic hand (Allegro Hand, Wonik Robotics), for precise manipulation of various objects in practical situations (Figure 7(c), Supplementary Video 6). We carried out three different tasks that require dexterous and precise manipulation. The first was manipulation of a ball on a table, repositioning the ball to a desired location and grasping it. The next task is to control the robotic hand to manipulate a mouse wheel, to scroll through pages of a document. This task required accurate control of the fingertip position so that the finger only rotates the mouse wheel without pressing it and activating a different mouse function. Finally, a volume control knob of an audio interface was manipulated by the robotic hand, which required unique sequences of multiple finger motions to rotate the cylindrical knob in both directions. The results shown in Supplementary Video 6 demonstrate that our glove system can be used to take full advantage of the dexterity of the human hand in various practical scenarios.”

- The use of FK to regulate hand pose sensing during hand gesture reconstruction is already a popular practice:

Liu et al., "A glove-based system for studying hand-object manipulation via joint pose and force sensing." IROS (2017)

Liu et al., "A new IMMU-based data glove for hand motion capture with optimized sensor layout." International Journal of Intelligent Robotics and Applications (2019)

We thank the reviewer for pointing out these details. As the reviewer has mentioned, the use of FK in reconstructing hand poses has already been done by many previous works including those referenced by the reviewer, and it was not our intention to present the use of FK method as a novelty of our research. Rather, we presented the robustness of the sensing glove in estimating arbitrary and even unconventional hand poses that go beyond the training data set, as one of the contributions of this glove system. We used the FK method to achieve this and the discussions on the FK method were

made to justify the use of stretch-based soft sensors and our hand pose reconstruction method. However, we realized the the Introduction may have been unclear in conveying these aspects of the glove and its novelty, and so we have edited the Introduction, as well as include the reference provided by the reviewer, as reproduced below:

“Furthermore, our system is able to reconstruct complex and unconventional poses taking advantage of the simple yet robust FK reconstruction method, making it suitable for various applications.”

...

“Finally, using the estimated kinematic parameters and joint motions, expressed in the form of the bone lengths and the joint angles, the system adopts the FK method to reconstruct the hand poses [39]. The FK method does not require a complex set of kinematic constraints often required for IK approaches [40] and the reconstruction is therefore not limited to the scope of the constraints or the training dataset. Taking advantage of this simplicity and efficiency, we demonstrate that our system is able to successfully reproduce the entire range of motions of the fingers, including various degrees of thumb oppositions, complex interactions, and unconventional poses of the fingers through multiple applications.”

Reviewer #3

---WORK SUMMARY---

This work presents a novel data glove that allows both anthropometric hand measurement and joint kinematics recording of thumb, index and middle fingers. The design and technical validation of the glove are described. This work is innovative and may be valuable for researchers devoted to hand kinematics, wearable systems or robotics.

---REVIEW SUMMARY---

This study addresses an important issue present when recording hand kinematics. A common practice when analysing hand posture is using the same kinematic model for all data collected. This approach can be useful in applications such as ergonomics or hand functional evaluation, but when the fingertip position is crucial, kinematic hand models should ideally include user hand anthropometric parameters. While these data can be obtained using optical motion capture systems, their use is sometimes cumbersome, with a very time-consuming setup and data processing (I suffered it!). Furthermore, in recordings involving product manipulation, marker occlusions may appear in optical systems. For these purposes, data gloves are the most appropriate systems, but they commonly provide data only of hand joints, or entire hand posture in space. This work presents a novel data glove that allows recording hand kinematics along with hand anthropometric measurements, being of significance for the researchers devoted to hand kinematics, among other fields. The main content of the work is relevant and well structured. The methodology is well described in detail, and does not have any flaw. Minor revisions have been proposed, but none of them questioning experimental phase or validation performed. The use of English is correct.

We sincerely appreciate your recognition of our efforts and all the constructive feedback you have provided. We have edited the manuscript to reflect your comments, and we believe they have further improved our paper.

---DETAILED REVIEW---

We appreciate your constructive feedback, and we believe they really helped improve the paper.

-TITLE-

The title clearly reflects the content of the work.

-ABSTRACT-

- *The abstract should somehow mention that the glove only measures thumb, index and middle fingers.*

As the reviewer has commented, we also agree the manuscript could benefit from a clarification on the specific fingers the proposed sensing glove directly measures, and so we have added the explanation as follows:

“The soft sensing glove is designed to easily stretch and to be one-size-fits-all, **both measuring** the size of the hand **and estimating the** finger joint motions **of the thumb, index, and middle fingers.**”

- *11: The word “dynamic” is a more general term that commonly involves also forces, mass, etc. It is used throughout the whole manuscript. In this specific case, maybe better just say “estimates finger joint motions” or “estimates finger joint kinematics”. Even though, this sentence seems quite similar to the first sentence of the abstract and could be both merged.*

Thank you for the feedback. As we have responded to the previous comment, we have edited and merged the two sentences as below:

“The soft sensing glove is designed to easily stretch and to be one-size-fits-all, both measuring the size of the hand and estimating the finger joint motions of the thumb, index, and middle fingers.”

- 14-16: *The sentence is difficult to follow, I would suggest proposing an alternative phrasing.*

The sentence was rephrased for clarity and specific values of the glove’s accuracy was also added as follows:

“The glove system is capable of reconstructing arbitrary and even unconventional hand poses with superb accuracy and robustness, confirmed by evaluations on the estimation of bone lengths (mean error: 2.1 mm), joint angles (mean error: 4.16°), and fingertip positions (mean 3D error: 4.02 mm), and on overall hand pose reconstructions in various applications.”

- *The abstract could benefit of a sentence mentioning the potential applications of the glove or the fields of study that would be interested in this glove.*

We have also added a sentence at the end of the Abstract to provide specific potential applications of the glove:

“The proposed glove allows us to take advantage of the dexterity of the human hand with potential applications, including but not limited to teleoperation of anthropomorphic robot hands or surgical robots, virtual and augmented reality, and collection of human motion data.”

-INTRODUCTION-

- *The introduction presents appropriately the state of the art and evidences the necessity to develop the glove.*
- 25: *I miss some references of the mentioned applications.*

We have added references to support the claims made in this sentence:

“Tracking and reconstruction of the hand articulation is, therefore, a popular topic of research with numerous applications, including robotics [5], healthcare [6], gaming [7], virtual and augmented reality [8].”

- 28: *I suggest substituting the word “rudimentary” for another without such level of negative connotations.*

We have edited the phrase to eliminate any negative connotation:

“Many prior studies have demonstrated classification and reconstruction of hand poses [9, 10, 11, 12, 13] albeit within a limited scope, providing partial evaluations on the hand motion tracking involving only part of the full range of finger motion, restricted hand positions, or simple applications limited to a small number of general hand poses.”

- *The introduction should also mention that the presented glove only measures thumb, index and middle fingers.*

We have also added the explanation that the glove system measures only the thumb, index and middle fingers to the third paragraph of the Introduction:

“The glove measures the lengths and the motions of the thumb, the index, and the middle fingers. However, the same sensing mechanism may be extended to measure the rest of all five fingers that share the same kinematic structure.”

-RESULTS-

- 97: *The kinematic structures of the ring and little fingers resemble to those of index and middle fingers, but this does not mean that they could be estimated from index and middle fingers motion. In fact, they present different motion patterns when analysing hand kinematics during varied tasks. They only can be assumed similar in power grasps (cylindrical, lumbrical, spherical, etc), but not in precision grasps, where the main role is assumed by thumb, index and maybe middle finger. Therefore, I suggest specifying “since the kinematic structures of the ring...”. Even though, the fact that the behaviour of these two fingers is not studied should be mentioned as a limitation of the work and maybe proposed as a future work.*

Thank you for pointing out these details. As you noted, according to the hand model shown in Fig. 2 (b), although the kinematic structure of the ring and little finger joints are identical to those of the index and middle fingers, their motion cannot be estimated just from the index and middle fingers with high accuracy. In this work, we have only demonstrated sensing of the thumb, index, and middle fingers, and the sensing mechanism can be extended to measure motion of all five fingers if needed. In reconstructing the hand motions with the virtual hand model, the joint angles of the ring and little fingers were assumed to be the angles of the middle finger multiplied by the constants $\frac{2}{3}$ and $\frac{2}{5}$ respectively. This was to provide a more natural visualization, but we did not conduct any quantitative analysis of these two finger joint motions. We understand this model may not be sufficient to cover all hand motions including those mentioned by the reviewer, and it was not our intention to suggest our glove system is capable of accurately sensing all five finger motions. We agree this is a limitation of our glove system in its current state, and this may not have been clearly conveyed in the initial manuscript, so we have edited the first paragraph of Section II. Results (previously line 97), and subsections *Post-processing method for identifying hand anatomy and extracting joint angles* and *Real-time hand pose reconstruction*:

“Since the kinematic structure of the ring and the little finger joints resemble those of the index and the middle fingers, we only test the sensing performance for estimating the lengths and motions of the thumb, the index and the middle fingers to prove the concept of our study.”

“Here, the motion of the ring and little fingers were not directly measured by the motion capture system. Their joint angles were assumed to be two-thirds and two-fifths of the corresponding joint angles of the middle finger, respectively, based on the concept of kinematic synergies of hand [60,61] for visualization purposes.”

“Since the ring and the little fingers were not directly measured by the glove, their rotation angles were assumed as two-thirds and two-fifths of the middle finger joints, respectively, for visualization purposes. These two fingers were not included in the evaluation of the performance of the glove system, which is a limitation of the glove in its current state since other variations of precision grasps may require independent motions of each finger for example. For a more comprehensive estimation of the full hand motion, the system can be easily extended using additional sensors and the same post-processing method, which we pose as future work.”

- 103: *Joint abbreviations used in figure 1 are missing in the caption or main text (specify CMC for carpometacarpal, MCP for metacarpophalangeal, etc.).*

We have added descriptions of the joint abbreviations to the legend of Fig. 1:

“Abbreviations: M-middle, I-index, T-thumb, PIP-proximal interphalangeal, IP-interphalangeal, MCP-metacarpophalangeal, CMC-carpometacarpal.”

- 103: *Does the index-middle abduction sensor hinder motion? Can the index finger fully cover its MCP flexion range while middle MCP remains at neutral position? From the supplementary material videos it cannot be entirely appreciated. It is just a personal concern.*

We acknowledged the reviewer’s concern with the abduction sensor of the index-middle finger hindering the motions of the MCP joints, and so have provided an additional video (Supplementary Video 1) showing various hand motions that cover the entire range of motion of each finger joint, including those pointed out by the reviewer. While the sensor does exert some load to the motion of the hand, the soft, stretchable silicone material as well as our implementation of a multi-thickness substrate structure allows this load to be minimized and does not obstruct the independent motions of the index and middle finger’s MCP joints. The particular hand pose mentioned by the reviewer had already been performed during the real-time hand pose reconstruction in Supplementary Video 2, and Figure 2 (b). However, to further show the natural and unrestricted motions of the hand, we have provided the additional Supplementary Video 1 where we show the movements of the hand while wearing our sensing glove in comparison to a bare hand.

In addition, we have also added more details about the mechanical characteristics of the sensing layer, in the subsection *Soft and sensitive liquid metal-based sensor* as reproduced below:

“The top sensing layer is composed of a silicone substrate (Ecoflex 00-30, Smooth-On, 100% modulus: 69 kPa, elongation at break: 900%) embedded with nine traces of eutectic Gallium–Indium (eGaIn), a room-temperature liquid metal [44].”

- 169: *The bone abbreviations are missing (DP, MP, PP and MB).*

Like the joint abbreviations, we have added the description of the abbreviations to the caption of Figure 2 as follows:

“Abbreviations: DP-distal phalanx, MP-middle phalanx, PP-proximal phalanx, MB-metacarpal bone.”

- 206-211: *Symbols “?” in equations.*
- 261-268-269-274-275: *Symbols “?” in equations.*

It appears all blank spaces in the equation have resulted in the “?” symbols during the conversion to the PDF file to the reviewer. This has been fixed in the revised manuscript.

- 223: *I would also comment here the assumption considered to represent ring and little fingers pose. Was it also assumed as two thirds of the middle finger joints?*

As responded above (comment for line 97), The joint angles of the ring and little fingers were each assumed to be two-thirds and two-fifth of the corresponding joint angles of the middle finger respectively, which we have added to the manuscript:

“Here, the motion of the ring and little fingers were not directly measured by the motion capture system. Their joint angles were assumed to be two-thirds and two-fifths of the corresponding joint angles of the middle finger, respectively, based on the concept of kinematic synergies of hand grasps [60,61] for visualization purposes.”

- 246: *Add to the caption something like: “Bone abbreviations as in Figure 2”.*

We agree this additional explanation could help clarify the bone abbreviations, so we have added the following description to the caption of Table 1:

“Bone abbreviations as in Figure 2.”

- 298-299: Maybe “covering the full range of motion of hand joints”.

We have reflected the suggestions to the manuscript as follows:

“During the experiment, the subjects performed diverse and unconstrained hand motions that utilized the full range of motion of each finger joint.”

- 379-404: *The videos are well prepared and clearly illustrate the applications tested, good job.*

Thank you, we truly appreciate your comments.

-METHODS-

The methods section presents all the analyses and procedures in detail and no important information is apparently missing. It is supported by the supplementary material, which provides details about the characteristics of the materials used, data collected during the several validation studies or key frames from the test applications videos.

- *In this section start to appear the abbreviations missing in the results section. Some of them should be defined there, in results section, to ease the manuscript readability.*

Thank you for the comments. As you have mentioned in your comments above for the Results section (for lines 103, 160), the descriptions of the abbreviations were added to the captions of Figs. 1 and 2.

-ADDITIONAL SUB-SECTION-

- *This manuscript could highlight the main strengths of the work presented by adding a sub-section remarking the main advantages of the glove designed regarding other recording methods, its applicability in several research fields and its limitations. Furthermore, this section could outline the future works, such as different angle estimation approaches (as mentioned in line 281) or studying also ring and little fingers. Maybe this could be a final sub-section in the results section. Something like “novelty, applicability and limitations”. It is just a suggestion, not a requirement.*

Thank you for the thoughtful suggestion. We agree the paper could benefit from an additional subsection that analyzes the novelty and limitations of our glove system in depth. So we have added a subsection at the end of the Results section, that compares our work to other previous work as well as commercial hand tracking systems, and discusses our work’s contributions, limitations, and future works. Table S5 was also provided to summarize the comparison of our glove system to others in various criterias. The edits are as follows:

“Contributions, limitations, and future works

Hand tracking systems have been widely studied and sometimes commercialized, but each with its own limitations in capturing the full dexterity of the human hand. In this work, we presented the contributions to this field of research, focusing on *accurate* estimation of the hand kinematic parameters and the hand motions, and also on *robust* reconstruction of various and even complex hand poses. We developed a wearable glove system, whose accuracy and robustness were improved by utilizing 1) a forward kinematic model tailored to the user’s specific anatomical structure, 2) diverse and unrestricted hand motion data acquired through a custom-designed motion capture apparatus and postprocessing, and 3) a compact stretchable sensing mechanism that simultaneously measures the kinematic parameters

and the joint motions. Through various quantitative evaluations (i.e., the estimations of the bone length, the joint angle, and the fingertip position), qualitative evaluations (i.e. the reconstructed hand poses and the Kapandji test), and practical applications (i.e., reconstruction of dexterous hand motions in virtual and real-world situations), we proved the distinguished performance of our hand tracking technology. Based on the contributions presented above and other criteria, we present a comparison of our technology to previous works and commercial products in Table S5, *Supplementary Information*.

Our technology also has a few limitations. Compared to commercial products, there is room for further improvements regarding the processing speed, the convenient user interface, and machine washability. In addition, we only demonstrated the sensing performance of three fingers since the ring and the little fingers have the same kinematic structure as the index and the middle fingers. While the same sensing mechanism and post processing method can be extended to the tracking of the ring and the little fingers, this would have to overcome the increased processing cost and also require a more extensive training dataset. As was mentioned in the subsection of *Real time refinement of bone lengths using signal distributions*, the likelihood model for the sensor signals should be verified and adjusted based on a larger dataset. Moreover, the joint angle estimation accuracy may be further improved by adopting a nonlinear sensor model. Finally, our wearable glove can also benefit from implementing haptic feedback to provide a more immersive and intuitive control interface for robot teleoperation.”

Reviewer #4

- *The authors propose an interesting system for finger tracking, but some of the declared advantages are not demonstrated, and the declared advantages with respect to other technologies are not quantified. Nevertheless, with major revisions this work can achieve some interest for the readers.*

Thank you for the constructive feedback. We have read through and reflected your comments and suggestions in the revised version of the manuscript, with details provided below.

I. Introduction

- *Q1: “rudimentary results” is a too generic sentence. Please provide justification/details.*

We had used the term ‘rudimentary’ to suggest many applications performed by these hand motion estimation systems or the evaluation conducted on them were limited to simple or very restricted hand gestures, a small portion of the finger joints’s full range of motion, or easy grasping applications. We agree the former phrase may have been too generic, and so we have edited the sentence as reproduced below:

“Many prior studies have demonstrated classification and reconstruction of hand poses [9,10,11,12,13], albeit within a limited scope, **providing partial evaluations on the hand motion tracking involving only part of the full range of finger motion, restricted hand positions, or simple applications limited to a small number of general hand poses.**”

- *Q2: Similarly, it is convenient to furnish some numbers/order of magnitude regarding the affirmed “precise localization of the fingertips and detailed posture of the hand”.*

As you have commented, we have edited the phrase to include more details on the required specifications as follows:

“Such performances may not be sufficient for applications, such as teleoperation of surgical robots [14], control of anthropomorphic robot hands [15,16], and clinical analysis [17], which demand precise localization of the fingertips **(e.g. 1 mm fiducial localization error of da Vinci system [18])** and detailed posture of the hand **(e.g. Kapandji score, a clinical assessment of the thumb opposition on a scale of 0 to 10 [19]).**”

- *Q3: “strain/bending sensors, encoders, or inertial measurement units (IMUs) ... vision or magnetic sensors” are the mostly adopted but not exclusive, and other approaches have been reported.*

Thank you for pointing out some of the works that we may have missed in our literature review. As your references have shown, a number of novel approaches have been made to track or distinguish motions of the hand. Therefore, we have also discussed these works in the second paragraph of the Introduction as reproduced below:

“To quantify the joint motion, prior studies have employed strain/bending sensors, encoders, or inertial measurement units (IMUs) to directly measure the joint angles or the bone rotations [21,22,23]. **Others have also utilized Triboelectric nanogenerators to detect charge generation with hand motions, or actuating-sensing bone conduction methods to analyze the mechanical waves traveling through the bones [24,25].** As a counterpart, vision or magnetic sensors have been used to track specific features of the hand, such as fingertips [11,26].”

- *Q4: “these methods of measurements require highly controlled environmental setups” is true for vision-based methods only. It is not true, for instance with the adoption of “strain/bending sensors”*

We thank you for your detailed comments. We realize the initial version of the manuscript was unclear as to which methods were regarded to be affected by the limitations explained in the sentence. So we have clarified our statement as follows:

“To identify the kinematic structures, vision-based techniques [27,28] have been widely used. However, these methods require highly controlled setups and are susceptible to the orientation of the cameras and occlusions, limiting the wearer’s free hand motion in various environments. Other studies have manually measured the bone lengths [29,30] but the result of this method can be highly dependent on the expertise of the examiner. A more simplified approach has also been adopted by applying the average bone lengths to all users rather than selecting personalized parameters [16], which however resulted in large errors in FK or provide multiple or no solution in IK [31,32].”

- Q5: “limiting the wearer’s free hand motions” and “this inevitably results in bulky and restrictive systems with reduced wearability” are not issues with not-mentioned technologies, such as bone conduction-based system (see: Saggio, G., Santoro, A. S., Errico, V., Caon, M., Leoni, A., Ferri, G., & Stornelli, V. (2021). A novel actuating–sensing bone conduction-based system for active hand pose sensing and material densities evaluation through hand touch. *IEEE Transactions on Instrumentation and Measurement*, 70, 1-7.).

Thank you for pointing out these works, which we may have missed in our first manuscript. We agree new approaches have been made in attempts to simplify the complexity of the hardware and enhance wearability, one example being the reference provided by the reviewer. However, we believe there are major differences in the contribution of this work and our research, as we have focused on providing highly accurate estimations of the individual finger joint angles, fingertip position, and finger bone length to reconstruct the hand poses, while the focus of the bone-conduction sensing method lies mostly on discriminating 3-5 hand poses between clenching a fist and opening the hand. So we have acknowledged these works as reproduced below:

“... this approach often results in bulky and restrictive systems with reduced wearability [35]. There have also been attempts to reduce the complexity of the hardware and enhance the wearability of the hand motion sensing system, focusing on differentiating a limited number of hand poses rather than precise estimations of the hand configuration parameters nor its reconstruction [25].”

- Q7: how overcoming “limiting the wearer’s free hand motions” is gained with your solutions? It is not mentioned in the following

We understand that the contents of the submitted manuscript and supplementary materials were not sufficient in demonstrating that our system does not limit the wearer’s hand motion. We have designed the textile interface using textiles of various stretchability, and fabricated the substrate of the sensing layer using soft stretchable silicone material, with details provided in the Supplementary Information. Furthermore, by adopting the multi-stiffness matrix structure in the sensor, we have further reduced the stiffness of the portions of the sensor that are subject to large strains. To clearly demonstrate that the sensor does not restrict natural motions of the wearer’s hand, we have added an additional supplementary video (Supplementary Video 1) depicting the process of anchoring the soft sensing layer onto the textile glove interface, and performing various hand motions that cover the full range of motions of each finger, in comparison to a bare hand.

In addition, we have also added more details about the mechanical characteristics of the sensing layer, in the subsection *Soft and sensitive liquid metal-based sensor* as reproduced below:

“The top sensing layer is composed of a silicone substrate (Ecoflex 00-30, Smooth-On, 100% modulus: 69 kPa, elongation at break: 900%) embedded with nine traces of eutectic Gallium–Indium (eGaIn), a room-temperature liquid metal [44].”

- Q8: “we propose a wearable glove ... with high accuracy”

Unfortunately, we are not quite sure what the reviewer intended to convey in this comment.

- Q9: “our system not only demonstrates a compact design” with respect to what other systems?

We had used the term 'compact' to mean our glove system takes advantage of the inherent properties of a stretch-based soft sensor to measure both the wearer's hand kinematic structure (finger bone lengths) as well as the motions of the finger (joint angle) without any additional sensors. Alternative methods to measuring the joint angle and finger lengths were discussed in the second paragraph of the Introduction, but these approaches presented more complex hardware systems, limited user environment or wearability, and larger errors when compared to our proposed system. To clarify what was meant in this sentence, we have made connection to the previous sentence as reproduced below:

“To the best of our knowledge, this is the first system capable of estimating both the kinematic parameters and the joint motions using a single proprioceptive sensing mechanism. **This compact design also demonstrates a consistent performance, unaffected by various factors, including position, orientation, or environmental conditions, making itself a truly wearable system.**”

II. Results

- Q10: “accurate and robust estimation” are claimed, but not mentioned in terms of what: reproducibility? Replicability? Sensitivity? Precision? ..
- Q17: You report “estimation errors” (Table 1), but you claim “accuracy” that must be addressed in terms of repeatability, reproducibility and reliability too, please add these missing parts, referring to a number of published papers addressing these concerns since Wise (Wise, S., Gardner, W., Sabelman, E., Valainis, E., Wong, Y., Glass, K., ... & Rosen, J. M. (1990). Evaluation of a fiber optic glove for se-automated goniometric measurements. *J Rehabil Res Dev*, 27(4).)

We believe both comments (Q10 & Q17) could be answered with the following response.

About *accuracy*: In this research, the term accuracy refers to the level of the error in finger bone length, joint angle, and fingertip position estimations, which was as small as 2.1 mm, 3.91°, and 3.24 mm, respectively with our sensing glove system. These results are summarized in Table 1, 2, and 3, with details in the testing procedures described in subsections *Initial estimation of bone lengths using initial sensor signal*, *Real time refinement of bone lengths using signal distributions*, *Real time estimation of joint angles*, *Fingertip position estimation*, and the *Supplementary Information*. Similarly, the term ‘accuracy’ has also been used to indicate the level of errors in the angular measurement in the finger joints in the reference provided by the reviewer.

About *robustness*: In this paper, the term ‘robustness’ does not represent the mechanical durability of the sensing glove. Rather, it refers to the capability of the glove system to sense and reconstruct arbitrary hand poses, not limited to the training dataset or conventional hand poses. It also means the performance of the glove is not affected by kinematic constraints often utilized with IK methods, or environmental constraints often found in vision based systems. This is possible because we use the FK method in reconstructing the hand poses, as was explained in the subsection *Real-time hand pose reconstruction* and also demonstrated through *Application 2: Shadowgraphy*, and *Supplementary Video 5*: “Next, to prove the robustness of our system in complex and unconventional poses, we demonstrated virtual shadowgraphy. ... Therefore, it is a good example that can highlight the robustness of our FK method that can accurately reconstruct even uncommon and unnatural hand postures using accurate finger bone lengths and joint angles estimations.” The term ‘robustness’ was also used in this manner in other researches regarding hand pose estimation, one example being: Lee et al., "Visual-inertial hand motion tracking with robustness against occlusion, interference, and contact." *Science Robotics* (2021).

To avoid further confusions with the terms, accuracy and robustness, we elaborated the terms when they were first used in our manuscript as:

“Therefore, there **still exists** a significant demand for a hand tracking system that overcomes these limitations in terms of accuracy (i.e., **hand reconstruction error [36]**), robustness (i.e., **ability to reconstruct arbitrary poses [35]**), and wearability (**physical comfort and compact form factor**).”

and we have also specifically presented the quantitative values regarding the accuracy of our glove system in the Abstract as reproduced below:

“**The glove system is capable of reconstructing arbitrary and even unconventional hand poses with superb accuracy and robustness, confirmed by evaluations on the estimation of bone lengths (mean error: 2.1 mm), joint angles (mean error: 4.16°), and fingertip positions (mean 3D error: 4.02 mm), and on overall hand pose reconstructions in various applications.**”

About *reproducibility*: We have provided experimental results regarding the reproducibility of the wearable sensing glove in the subsection *Initial estimation of bone lengths using initial sensor signal* and *Figure S2, Supplementary Information*, and the testing procedures in *Methods-Testing procedure and characterization method for initial estimation of bone lengths using initial sensor signal*, where subjects were instructed to put on and remove the sensing layer five times during which the sensor signal was alternately collected. Also, since our proposed glove system is intended to estimate and reconstruct the hand in real time during dynamic hand motions rather than static cases, we did not perform the repeatability test as was done by Wise. S in the provided reference. Instead, quantitative evaluations of the joint angle and fingertip position estimation were made while performing dynamic hand motions over a certain time period, and the errors were averaged over the test time.

About *sensitivity*: In the previously submitted manuscript, we had not provided specific quantities regarding the sensitivity of the sensing glove. We thank the reviewer for pointing this out, and we have added details on the sensitivity along with other material properties of the silicone material used for the sensor substrate. We have also provided more details on sensitivity in our response to Q11 and Q15.

- Q11: “*a liquid-metal soft sensor layer characterized by its high sensitivity and flexibility*”: what does it mean “*high sensitivity*”? “*High*” is an adjective not a measurement and “*flexibility*” is not defined with a number.

Thank you for your comment. We understand that our previous statements have not been specific enough regarding the sensitivity and flexibility of the sensor. We had originally intended to highlight the multi-stiffness substrate structure of our liquid metal sensor, which increases the sensitivity of conventional single-material sensors. This was adopted from our previous work (Myungsun Park, Taejun Park, and Yong-Lae Park. "Parametric analysis of multi-material soft sensor structures for enhanced strain sensitivity." *Extreme Mechanics Letters* 60 (2023): 101983), as was discussed in *Results-Soft and sensitive liquid metal-based sensor*. We have revised the manuscript to reflect this more clearly, as well as provide specific quantities regarding the sensitivity and the material property of the sensor substrate in the subsection *Soft and sensitive liquid metal-based sensor*, as reproduced below:

“The top sensing layer is composed of a silicone substrate (Ecoflex 00-30, Smooth-On, **100% modulus: 69 kPa, elongation at break: 900%**) embedded with nine traces of eutectic Gallium–Indium (eGaIn), a room-temperature liquid metal [44].”

“The mechanical structure of the sensing layer was designed to **increase the resolution of liquid-metal soft strain sensors**, resulting in accurate measurement of the bone lengths and the joint angle.”

“**These sensors are also known for their reliability in prolonged and repeated loading cycles, which overcomes the structural limitation often faced by soft, stretchable sensors. We thus employed this approach**, creating the stiffness variations by altering the thickness of the substrate with a single material, as shown in Figure 1(b). This method **not only** enhanced **the structural integrity** of the sensor **but** also **improved** the sensitivity (**gauge factor: 3.4**).”

- Q12: “to accommodate hands of all sizes”: hands differ not only in length but is diameter too. How this issue is considered?

We understand your concerns regarding the various hand sizes, especially the varying palm and finger diameter. We have designed both the textile glove interface and the soft sensing layer to accommodate various finger lengths as well as varying finger and palm diameters. This was achieved by using stretchable silicone materials as the substrate of the soft sensing layer, and also using a unique combination and layout of fabric materials with varying stretchability for the textile glove interface (Figure 1 (d), and properties of textiles reported in *Supplementary Information*). So the textile glove was designed to stretch both axially and radially when worn on larger hand sizes. In addition, a detachable and size-adjustable hook-and-loop ring is fastened in the middle of each finger bone to provide anchor points for the soft sensing layer, which can also accommodate larger finger diameters. You may refer to *Methods-Fabric selection and patterning in custom textile glove interface*, which explains the design of the textile glove interface in more detail.

- Q13: “Since the kinematics of the ring and the pinky fingers closely resemble those of the index and the middle fingers”: in attached videos, the pinky finger does not move “closely resemble”, please clarify. Moreover your sentence is not always true. In addition, what does “closely resemble” mean? In terms of what?

Thank you for pointing out these details. The phrase “closely resemble” refers to the kinematic structure of the ring and little finger joints that are identical to those of the index and middle finger, as shown in Fig. 2 (b). To avoid any confusion, we have replaced the term “kinematics” with “kinematic structures”. Please note that in this work, we have only demonstrated sensing of the thumb, index, and middle fingers to prove the concept of our sensing glove, and the sensing mechanism can easily be extended to measure motion of all five fingers if needed. In reconstructing the hand motions with the virtual hand model, the joint angles of the ring and little fingers were assumed to be the angles of the middle finger multiplied by the constants $\frac{2}{3}$ and $\frac{2}{5}$ respectively. This was to provide a more natural visualization, but we did not conduct any quantitative analysis of these two finger joint motions. We understand this model may not be sufficient to cover all hand motions, two-fingered precision grasps being one example, and it was not our intention to suggest our glove system is capable of sensing all five finger motions. We agree this is a limitation of our glove system in its current state, and this may not have been clearly conveyed in the initial manuscript, so we have edited the first paragraph of the Results section, and subsections *Post-processing method for identifying hand anatomy and extracting joint angles* and *Real-time hand pose reconstruction* as reproduced below:

“Since the kinematic structure of the ring and the little finger joints resemble those of the index and the middle fingers, we only test the sensing performance for estimating the lengths and motions of the thumb, the index and the middle fingers to prove the concept of our study.”

“Here, the motion of the ring and little fingers were not directly measured by the motion capture system. Their joint angles were assumed to be two-thirds and two-fifths of the corresponding joint angles of the middle finger, respectively, based on the concept of kinematic synergies of hand grasps [60,61] for visualization purposes.”

“Since the ring and the little fingers were not directly measured by the glove, their rotation angles were assumed as two-thirds and two-fifths of the middle finger joints, respectively, for visualization purposes. These two fingers were not included in the evaluation of the performance of the glove system, which is a limitation of the glove in its current state since other variations of precision grasps may require independent motions of each finger for example. For a more comprehensive estimation of the full hand motion, the system can be easily extended using additional sensors and the same post-processing method, which we pose as future work.”

- Q14: “large deformations while exerting minimal force”: how “large” and how “minimal”?

Thank you for your comment. We realize the phrase “large deformations while exerting minimal force” may not have been technical enough to convey the characteristics of our sensing layer, so we have removed the phrase. Instead, to quantify ‘how large’ the deformation is and ‘how minimal’ the force is, we have added the 100% modulus and the maximum elongation of the material used to fabricate the sensing layer in the subsection *Soft and sensitive liquid metal-based sensor*.

“The top sensing layer is composed of a silicone substrate (Ecoflex 00-30, Smooth-On, 100% modulus: 69 kPa, elongation at break: 900%) embedded with nine traces of eutectic Gallium–Indium (eGaIn), a room-temperature liquid metal [44]. The soft and stretchable substrate can accommodate a wide range of movements and conform to irregular contours of the human hand [45,46].”

- Q15: “*The mechanical structure of the sensing layer was designed to achieve the increased resolution, resulting in a more accurate measurement of the bone lengths and the joint angle*”: “*more accurate*” with respect to what?

As responded in Q11, our sensing layer, composed of silicone and liquid metal (LM) sensing channels, demonstrates enhanced sensitivity and resolution compared to other more common LM-based stretch sensors due to its multi-stiffness matrix structure, adopted from our previous work (Myungsun Park, Taejun Park, and Yong-Lae Park. "Parametric analysis of multi-material soft sensor structures for enhanced strain sensitivity." *Extreme Mechanics Letters* 60 (2023): 101983). However, we agree the submitted manuscript was unclear as to what the resolution and accuracy were compared to, and so we have edited the sentence as reproduced below:

“The mechanical structure of the sensing layer was designed to **increase the resolution of liquid-metal soft strain sensors**, resulting in accurate measurement of the bone lengths and the joint angle.”

- Q16: “. *By securing the layer onto the middle of each finger bone, each sensor was positioned directly over each finger joint*”: did you consider possible misalignment issues? (see: Saggio, G. (2014). *A novel array of flex sensors for a goniometric glove. Sensors and Actuators A: Physical*, 205, 119-125.).

As you have pointed out, and as your reference has demonstrated, we agree that the misalignment issue and proper electrical wiring is very critical to the performance of a sensing glove. We tackled this problem using the hook-and-loop (velcro) based anchoring system that links the sensing layer and the textile glove interface. These velcro rings are firmly fastened onto the middle of each finger bone, and act as fixed points with no rotation or displacement even during hand motions. The anchor ring is also designed to be detachable and size-adjustable according to the user’s finger diameter to ensure the stable fixture. These details have been explained in subsections *Soft and sensitive liquid metal-based sensor*, and *Soft and sensitive liquid metal-based sensor*. The sensing channels are directly printed onto a flat flexible cable, connected to the wireless circuit board watch through a Molex connector, ensuring stable wiring and avoiding potential error or noise.

We have also added Supplementary Video 1 to show stable anchoring and consistent alignment of the sensor during various hand motions.

REVIEWERS' COMMENTS

Reviewer #1 (Remarks to the Author):

Thank you for revising the manuscript. It has clarified certain doubts and pointed out their novelty, albeit not very significant but sufficient. With some skepticism I recommend this manuscript to be published as it is.

Reviewer #3 (Remarks to the Author):

The authors have effectively addressed all raised concerns and implemented all the changes suggested by the four reviewers, including those based on my feedback. Specifically, the comments and changes I recommended have been satisfactorily addressed. Furthermore, in the current version the authors have not only improved the manuscript itself but have also included Supplementary Material Videos to enhance the clarity and comprehensibility of the content.

Consequently, with regard to the aspects within my field of expertise that I could assess, I recommend accepting the manuscript in the present form.

(After being asked to comment on the remaining concerns of Reviewer #2):

I revised again the manuscript and the responses to reviewer's #2 comments. Considering the four main bullet points in which authors divided the answers to the reviewer, here are my thoughts:

1. The response of the authors is clear and supported by previous studies by the authors. Nevertheless, I recommend adding to the manuscript something more explicit regarding the testing performed in the previous work. Something like "These sensors are also known for their reliability in prolonged and repeated loading cycles, as demonstrated in previous studies [48], which overcomes the structural limitation often faced by soft, stretchable sensors."
2. Authors added the standardised Kapandji test, but modified according to the fingers recorded in this glove, which makes sense. I consider the comment to be appropriately addressed.
3. Authors demonstrated the applicability of the glove by performing further experiments that included teleoperation of the Allegro Hand in fine manipulation tasks, and adding Supplementary material to illustrate it.
4. Authors emphasized in the main text that the method used has been previously used by other authors. They cited previous works and also used expressions such as "the system adopts the FK reconstruction method", which somehow states that this method is not novel.

Reviewer #4 (Remarks to the Author):

The authors addressed my main concerns. Now the paper can be accepted for publication